



# Mineralogy and geochemistry of Asian dust: Dependence on migration path, fractionation, and reactions with polluted air

Gi Young Jeong*

Department of Earth and Environmental Sciences, Andong National University, Andong 36729, Republic of Korea

*Correspondence to*: Gi Young Jeong (jearth@anu.ac.kr)

**Abstract**. Mineralogical and geochemical data are essential for estimating the effects of long-range transport of Asian dust on the atmosphere, biosphere, cryosphere, and pedosphere. However, consistent long-term data sets of dust samples are rare. This study analyzed 25 samples collected during 14 Asian dust events occurring between 2005 and 2018 on the Korean Peninsula, and compares them to 34 soil samples (< 20 μm) obtained from the Mongolian Gobi Desert, which is a major source of Asian dust. The mineralogical and geochemical characteristics of Asian dust were consistent with those of fine source soils in general. In dust, clay minerals were most abundant, followed by quartz, plagioclase, K-feldspar, calcite, and gypsum. The trace element contents were influenced by mixing of dust with polluted air and fractionation of rare earth elements. Time-series analyses of the geochemical data of dust, combined with satellite remote sensing images, showed a significant increase of Ca content in the dust crossing the Chinese Loess Plateau and the sandy deserts of northern China. Calcareous sediments in the sandy deserts and pedogenic calcite-rich loess are probable sources of additional Ca. Dust-laden air migrating toward Korea mixes with polluted air over East Asia. Gypsum, a minor mineral in source soils, was formed by the reaction between calcite and pollutants. This study describes not only the representative properties of Asian dust, but also their variation according to the migration path, fractionation, and atmospheric reactions.

## 1 Introduction

Mineral dust blown from arid lands is transported to remote atmospheric, terrestrial, cryogenic, and marine environments, contributing to the circulation of earth materials (Martin and Fitzwater, 1988; Dentener et al., 1996; Biscaye et al., 1997; Jickells et al., 2005; Mahowald and Kiehl, 2003; Zdanowicz et al., 2007; Formenti et al., 2011;



Jeong et al., 2013; Jeong et al., 2014; Serno et al., 2014). Asian dust is a major mineral dust that are the subject of ongoing interdisciplinary research. The mineral grains of dust interact with atmospheric gases and pollutants (Dentener et al., 1996; Krueger et al., 2004; Laskin et al., 2005; Matsuki et al., 2005), which affects the bioavailability of inorganic micronutrients in remote ecosystems (Meskhidze et al., 2005; Takahashi et al., 2011). The interaction of dust particles with solar and Earth radiation influences the regional energy balance (Forster et al., 2007). The long-term deposition of dust particles on the Loess Plateau, the North Pacific Ocean, and Arctic ice sheets provides a record of paleoclimatic changes (Liu et al., 1988; An et al., 1991; Porter, 2001; Pettke et al., 2000; Bory et al., 2002; Hyeong et al., 2005; Jeong et al., 2008, 2011, 2013). Asian dust transported over long distances is an important constituent of some soils in Korea and Japan (Bautista-Tulin and Inoue, 1997; Jo et al., 2019). Iron-bearing dust transported to remote oceans has received much attention for its possible role in phytoplankton bloom and carbon dioxide levels (Jickells et al., 2005; Johnson and Meskhidze, 2013). Mineralogical and geochemical analysis of dust extracted from pelagic sediments of the North Pacific provided a basis for determining sediment provenance and paleoenvironmental changes (Olivarez et al., 1991; Nakai et al., 1993; Leinen et al., 1994; Rea, 1994; Rea et al., 1998; Pettke et al., 2000; Hyeong et al., 2005; Serno et al., 2014).

Mineralogical and geochemical properties of bulk samples provide a basis for interdisciplinary research on long-range transported Asian dust. The earth system models involving Asian dust could be improved by adopting reliable data on dust properties. However, the bulk properties of Asian dust are poorly known due to application of widely varying analytical procedures and sample weights that are typically very low and thus usually insufficient for analysis (Leinen et al., 1994; Kanayama et al., 2002; Shi et al., 2005; Zdanowicz et al., 2007; Jeong, 2008; Jeong et al., 2014). Furthermore, analyses of scattered sample sets of limited number are difficult to detect the long-term variation of dust properties. Thus, long-term data sets obtained using consistent analytical methods could provide not only general mineralogical and geochemical properties of Asian dust for earth system modeling, but also insights into the change of dust source, migration path, and chemical interactions. However, no such data have been reported to date. Mineralogical and geochemical data for Asian dust should be compared to equivalent data for the fine silt fraction in the source soils for investigating any fractionation and reaction during the long-range transport. The currently available mineralogical and geochemical data for source soils (Biscaye et al., 1997; Honda et al., 2004; Chen et al., 2007; Jeong, 2008; Maher et al., 2009; Ferrat et al., 2011, McGee et al., 2016) are insufficient; mineralogical data are particularly rare.

The purpose of this study is to determine the mineralogical and geochemical properties of bulk dust samples





collected over 14 years and compare them to equivalent data for source soils. Variations of the mineralogical and geochemical characteristics are discussed in relation to migration path, fractionation, and the interaction of Asian dust with atmospheric pollutants.

## 2 Samples

### 2.1 Asian dust

#### 2.1.1 Outbreak and migration of dust storms

The outbreak and migration of dust storms crossing Korean Peninsula were investigated for 14 Asian dust events using an aerosol index derived from data obtained by the Communication, Ocean, and Meteorological Satellite (COMS), which was launched on 27 June 2010 (National Meteorological Satellite Center, 2019). Pre-2011 dust events were tracked using the Infrared Difference Dust Index derived from data obtained by the Multi-functional Transport Satellite-1R (MTSAT-1R) (National Meteorological Satellite Center, 2019). Data for 2005 dust events were not available. Four satellite images were selected from the serial image set (1~0.5 h intervals) of each dust event to show 1) the extent of the dust outbreak (i.e., the point where the dust storm reached its maximum size without notable migration), 2) dust migration toward the Korean Peninsula, 3) dust crossing the Korean Peninsula, and 4) dust leaving Korean Peninsula toward the North Pacific Ocean. The dust region identified from the dust index images was drawn on a geographic map including the Gobi Desert, sandy deserts of northern China, and Loess Plateau. The outbreak and migration of each dust storm identified from satellite images are provided in Supplementary Fig. S1. Fig. 1 is a summary of the outbreak and migration of all the dust storms. The outbreak of dust storms was concentrated in the Gobi Desert and sandy deserts encompassing southern Mongolia and northern China (Fig. 1a). The 2013 dust storm occurred in the Loess Plateau as well as deserts (Supplementary Fig. S1). The dust storm migrated eastward and southeastward (Fig. 1b). The migration routes of dust storms toward the Korean Peninsula can be divided into two groups: 1) those crossing the Loess Plateau and 2) those making a detour around the north of Loess Plateau. Six dust storms (D3–6, D7, D10–11, D12, D19–20, and D21–23 in Table 1) crossed loess plateau (Supplementary Fig. S1). Dust-laden air parcels passing the Korean Peninsula are dispersed and



diluted progressively, and migrate eastward and northeastward toward the North Pacific Ocean (Figs. 1c–d).

### 2.1.2 Size distribution

Volume size distribution was measured with an optical particle counter (OPC) at dust monitoring station nearest to sampling site operated by Korea Meteorological Administration (2019). OPC data for pre-2010 dust events were not available. The volume size distributions revealed that the modal volume diameters of most dusts were between 2–5 μm with an average size of 4.6 μm (Fig. 2). The volume size distribution of very coarse dust for a 2012 dust event showed an almost monotonic increase toward larger sizes (Jeong et al., 2014). Zdanowicz et al. (2007) reported a modal volume diameter of 4 μm of Asian dust transported long distances in April 2001, (to the Yukon
Territory, Canada); larger particles (> 10 μm) were also found, indicating rapid trans-Pacific transport in the mid-troposphere. Serno et al. (2014) reported a particle-size mode of around 4 μm for eolian dust separated from deep-sea sediments of the subarctic North Pacific Ocean. It is remarkable that particle size of dust is uniform from the western margin of the North Pacific Ocean to the subarctic mountains of the North America.

### 2.1.3 Sampling

Twenty-five samples of Asian dust were collected from 2005 to 2018 at three sites, including Deokjeok Island off the western coast of Korea, Andong National University in Andong, and the Korea Institute of Science and Technology in Seoul (Fig. 3 and Table 1). Dust particles were collected on Whatman No. 1441–866 cellulose filter
paper; via Tisch Environmental and Thermo Scientific high-volume total suspended particulate (TSP) samplers installed on building roofs. Jeong (2008) reported on the mineralogical properties of eight dust samples collected during 3-year period (2003–2005). However, his samples were collected using $PM_{10}$ samplers; these exclude coarse particles, which are an important component of some dust events (Jeong et al., 2014). For some events, dust samples were collected at three sites, whereas the samples for other events were collected at one or two sites, depending on
the dust migration path. For short events (a few hours in duration) one sample was collected, while 2~4 time-series of samples were collected at one site during longer events (several days in duration). The mineralogical properties of three TSP samples reported by Jeong et al. (2014) were re-analyzed to ensure consistency in the analytical procedures.





## 2.2 Source soils

The 34 surface soils were sampled in the Mongolian Gobi Desert along a track of 1,700 km long in the region ca.
E100º~109º and N42º~46º(Fig. 3 and Table 1). The surface soils of Gobi Desert are not dominated by the sands
10  typical of sandy desert, instead being characterized by a mixture of pebbles, sand, silt, and clays, although sand
dunes are locally distributed. The bare ground, comprising loose silty soils with sparse vegetation and the dry beds
of ephemeral lakes, promotes the outbreak of dust plumes under strong winds caused by a cold front system, or by
a strong pressure gradient at the surface (Chun et al., 2001). About 1 kg of soil samples were taken from the surface
after coarse pebbles were removed. All the soil samples were in the naturally dry state at the sampling time.

## 3 Analytical Methods

### 3.1 Sample preparation

20  The cellulose filter papers were shredded into several pieces, each approximately $3 \times 5$ cm$^2$ in size, and subjected
to ultrasonic agitation, in methanol in a 250 mL glass beaker, to detach dust particles from the filter. Cellulose
fibers were removed from the dust suspension by passing through a 270 mesh sieve. The dust suspension was dried
on a clean glass plate, and then collected with a razor blade for X-ray diffraction (XRD) and chemical analysis.
Soil samples were passed through clean 2 mm sieve to remove pebbles. The 2-mm fraction soils was sieved under
25  dry conditions through a disposable nylon 20-μm sieve, using a Retsch sieve shaker. 2 g of the soil separate (< 20
μm) was mixed with ethanol and ground using a McCrone micronizing mill for 7 min with zirconia grinding
elements. The dust samples were not ground for XRD analyses to avoid mass loss during the milling, because the
dust samples were very small (< 300 mg) and already sufficiently fine for XRD analyses. However, one sample
collected during the coarse dust event was ground in an agate mortar.

### 3.2 XRD analyses



Since the weight of the dust samples was insufficient for conventional XRD analysis, XRD data were collected over a long time period (~12 h). Dust powders were loaded onto a small cavity (~ $7 \times 20$ mm$^2$) in a silicon plate for XRD analysis by side packing to minimize preferred orientation of mineral grains. The XRD analyses were performed using a Rigaku Ultima IV diffractometer at the Center for Scientific Instruments, Andong National University. The analytical conditions were as follows: counting time, 20 s per 0.03° step; 2θ, 3–65°; divergent slit, 2/3°; scatter slit, 2/3°; receiving slit, 0.15 mm; and Cu Kα radiation, 40 kV/30mA. The counting time was doubled for the samples of very low weight. Given their higher weight, the soil powders were loaded on the $20 \times 20$ mm$^2$ cavity by side packing, and analyzed at a scan speed of 0.25° per min. Mineral identification based on the XRD patterns was carried out with DIFFRAC.EVA software (Bruker AXS).

Twelve minerals (quartz, plagioclase, K-feldspar, illite, illite-smectite mixed layers, chlorite, kaolinite, amphibole, calcite, dolomite, gypsum, and halite) were quantified using SIROQUANT software (Sietronics Ltd.) with application of the Rietveld refinement technique. Background subtraction was performed carefully because samples enriched with poorly-crystalline clay minerals have high and unresolvable broad diffraction bands, particularly in the range of 20–40° 2θ. The low angle region (3–10°) was excluded from the refinement. After initial refinement, the cell parameters of chlorite, K-feldspar, albite, and calcite were refined to achieve the best fit between the observed and calculated XRD patterns. Although smectite is present as a minor clay mineral in source soils and dust (Jeong et al., 2008; Park and Jeong, 2016), it was excluded from refinement because the XRD patterns of randomly oriented bulk samples are not adequate for distinguishing small amounts of smectite from illite-smectite mixed-layer clay minerals. The refinement often showed that low crystalline illite is difficult to be reliably distinguished from illite-smectite mixed-layer minerals in dust. This was confirmed by transmission electron microscopy analyis of clay minerals (Jeong and Nousiainen, 2014; Jeong and Achterberg, 2014) and single-particle analysis using scanning electron microcopy (SEM) (Jeong et al., 2016). Thus, in this study, the illite-smectite mixed-layer minerals, illite, and smectite contents are summed and defined as illite-smectite series clay minerals (ISCMs), as in previous works (Jeong et al., 2016). Five dust samples collected in the last five years were independently quantified by single-particle analysis using SEM combined with energy dispersive X-ray spectroscopy (EDS) analysis, following the method described in the supplement of Jeong et al. (2016).

### 3.3 Geochemical analyses



The major and trace element contents of the soil separates were determined in Activation Laboratories (Ontario, Canada). Dust and soil samples were mixed with a flux of lithium metaborate and lithium tetraborate, and fused in a furnace. The melt was dissolved in a solution of 5% nitric acid. The solutions were run on a Varian Vista 735 inductively coupled plasma (ICP) emission spectrometer for major elements (Si, Al, Fe, Mg, Ti, Mn, Ca, Na, K,

and P) and several trace elements (Ba, Sc, Sr, V, Y, and Zr). The solutions also were run on a Perkin Elmer Sciex ELAN 9000 ICP mass spectrometer for the other trace elements. For the analysis of selected trace elements (Cu, Ni, Pb, S, and Zn), samples were digested to solutions with hydrofluoric, nitric, perchloric, and hydrochloric acids, and analyzed using a Varian Vista ICP emission spectrometer. Analytical quality was controlled by using USGS and CANMET certified reference materials for calibrations, internal standards, and duplicate analyses. Detection

limits of major elements were under 0.01%. Detection limits of trace elements were provided in Table 4. Loss on ignition was not measured in either the dust or soil samples. The dust samples prepared for chemical analyses ranged from 30 to 100 mg in weight. The Cu, Ni, Pb, and Zn contents were measured only for 11 samples of enough sample weights.

**4 Results**

**4.1 Mineralogy**

**4.1.1 Asian dust mineralogy**

The mineral compositions of Asian dust determined by XRD are presented in Table 2. Five dusts were independently quantified by SEM-EDS single-particle analyses. The XRD data are in good agreement with the SEM-EDS data, particularly the quartz content, total clay minerals, and ratio of plagioclase to K-feldspar. The SEM-EDS data support the reliability of the XRD quantifications (Table 2). However, the SEM-EDS analyses

somewhat underestimated the K-feldspar, amphibole, and gypsum contents. This underestimation was due to ambiguity in the interpretation of the EDS patterns of some of the dust particles that are normally present as mixtures of several minerals. For example, K-feldspar is difficult to recognize from the mixture particle of K-



feldspar and ISCMs because both K and Al are major constituents of two phases. Amphibole is difficult to distinguish unambiguously from the mixture particle of ISCMs and calcite. Distinguishing between calcite and gypsum in a mixture is also difficult. Thus, XRD quantification is probably more reliable than SEM-EDS quantification.

Clay minerals accounted for an average of 48% of the total mineral content. ISCMs were the major clay mineral (42% on average), followed by chlorite (4%) and kaolinite (2%) (Table 2). Although ISCMs also contained smectite, they were dominated by illite and illite-smectite mixed layers. A weak peak corresponding to expanded smectite was detected by XRD analysis of ethylene glycol treatment (Jeong, 2008; Park and Jeong, 2016). XRD analysis of clay minerals (< 2 μm) of the Chinese loess, which is a deposit of Asian dust, confirmed that clays are dominated by illite and illite-smectite mixed-layer minerals, with only minor amounts of smectite (Jeong et al., 2008).

The quartz content of dust samples was around 20%, and showed a roughly inverse relationship with the total clay mineral content ($R^2$=0.37). Samples of very coarse dust from a 2012 dust event (D10–11) (Fig. 2) showed the highest quartz and lowest clay mineral content (Table 2). The average feldspar (plagioclase and K-feldspar) content was 18%. The average ratio of plagioclase (12.4%) to K-feldspar (5.1%) content was 2.5. Amphibole was detected as a minor mineral (2% on average).

Carbonates and gypsum are important constituents of dust because of their reactivity and solubility. The average content of calcite was 5%, but varied widely between 0.5 and 11%. The average content of gypsum was 5%, but this also varied widely, between 0.2 and 18.3%. Dolomite was a minor component, being present in proportions of around 1%. A small quantity of halite was detected only in a dust sample collected in Deokjeok Island (D17) during the 2015 dust event. Although iron oxides (goethite and hematite) are minor minerals (~ 1–2%) responsible for the yellow-brown color of dust, they were not quantified due to their low crystallinity.

### 4.1.2 Temporal variation of Asian dust mineralogy

Fig. 4 shows the temporal variation of mineral content in the dust samples. The clay mineral content varied in the opposite direction to the quartz and feldspar contents. Quartz-rich dust (D11) was sampled during the coarse dust event. The quartz, feldspar, and clay mineral contents do not show any significant correlations with calcite and gypsum contents.





Three sets of intra-event dust samples (sets 1, 5, and 6 in Fig. 4) were collected at the same site. Set 1 samples showed little intra-event variation, while set 6 samples showed clear increases in clay and gypsum contents toward the end of the dust event, along with decreases in quartz, feldspar, and calcite contents. Set 5 samples showed a decrease of clay-mineral content, but with no notable change in quartz content.

The other sets of dust samples (sets 2–4 in Fig.4) were collected at different sites during the same dust event, and showed little spatial variation in mineral composition. However, in set 3, the gypsum content in samples from Deokjeok Island (D14) was far higher compared to samples from Andong (D13).

### 4.1.3 Soil mineralogy

The mineral compositions of surface soils (<20 μm) in the Mongolian Gobi Desert are presented in Supplementary Table S1. The mineral compositions varied among samples, probably according to the local geology. The total clay mineral content ranged widely among samples, from 25.3% (G18) to 67.4% (G34). This range was wider than that among the dust samples (33.8~59.1%). Clay minerals are dominated by ISCMs, followed by chlorite (4.3%) and kaolinite (3.0%). XRD analyses of clays (< 2 μm) treated with ethylene glycol revealed variation of the smectite content (Supplementary Fig. S2). Although some sample was enriched with smectite (sample G34), the XRD intensities of smectite were generally weak; illite and illite-smectite mixed layers tended to dominate ISCMs. The quartz content varied widely among samples, from 8.8% (G34) to 32.1% (G18), in the opposite direction to the clay mineral content. The average ratio of plagioclase (14.3%) to K-feldspar (5.5%) was approximately 2.6. Calcite contents were 9.5% on average. Soil samples G2, G19, and G28 exhibited calcite enrichment (> 20%). The calcite-rich samples are abundant in limestone pebbles (G2), and secondary calcite precipitates (G19 and G28). Gypsum was a minor component of source soils (0.6% on average), but was more abundant in samples G22 (5.6%) and G31 (3.2%) from dry lake beds, and in sample G25 (4.4%) from a dry river bed.

### 4.1.4 Comparison of mineralogy between Asian dust and soil samples

Mineral compositions are compared between the source soils and Asian dust samples in the box-whisker plots shown in Fig. 5. The plots show similar ranges of mineral contents between the Asian dust and source soil samples. Calcite and gypsum contents, however, differed between dust and soil samples. In source soils, gypsum was present





only in trace amounts (average content of 0.6%), while calcite was abundant (average content of 9.5%) (Supplementary Table S1, Fig. 5). In the dust samples, however, the average gypsum content was 5%, while the average calcite content was 5.1% (Table 2, Fig. 5).

## 4.2 Geochemistry

### 4.2.1 Comparison of geochemistry between Asian dust and soil samples

The major-element compositions of dust samples were recalculated on a volatile-free basis and are presented as metal wt% values (Table 3). Trace element compositions of dust samples are listed in Table 4. The major and trace element compositions of the source soil samples are listed in Supplementary Tables S2 and S3, respectively. The contents of major and trace elements of Asian dust and source soil samples were normalized by the average values for the upper continental crust (UCC) (Rudnick and Gao, 2003) and are presented in Figs. 6–8.

Major-element compositions of both the dust and soil samples did not coincide with the average UCC values (enrichment factor = 1). Si and Na were relatively depleted in dust and soils, compared with the average UCC (enrichment factor < 1), while Al, Fe, Mn, Mg, Ca, K, Ti, and P were relatively enriched in dust and soils (enrichment factor > 1) (Fig. 6). In general, the range of the major-element contents of Asian dust samples coincided with those of source soils (Fig. 6). However, Al, Fe, Mn, Mg, K, and P were slightly enriched in dust in comparison to soils. The ranges of Al, Fe, K, and Ti contents of dust samples were narrower than those of source soils. The ranges of Ca and Na contents were wide in both the dust and soil samples.

The UCC-normalized trace element composition data showed that dust samples were significantly enriched with Cu, Zn, Sn, Pb, and S relative to the source soil samples (Fig. 7). The dust samples were slightly enriched with Cr, Co, Ni, Sb, and Ba relative to the soil samples, while Sr, Y, Zr, Hf, and Ta were slightly depleted in dust (Fig. 7). Compared with the soil samples, the rare earth elements (REEs) of dust indicated systematic depletion of heavy REEs (Tb–Lu) relative to light REEs (La–Nd) (Fig. 8). The contents of V, Ga, Th, and U showed little difference between the dust and soil samples.

### 4.2.2 Temporal variation of Asian dust geochemistry





Time-series data of UCC-normalized major element contents are provided in Fig. 9. The Si, Al, Fe, and Ti contents showed little fluctuation within an enrichment factor of +/– 0.5. However, Ca, Mg, and Na contents exhibited large fluctuations over time.

Time-series data of UCC-normalized trace element contents showed that REEs and Zr exhibited small variations around an enrichment factor of 1, whereas Sn fluctuated significantly (Fig. 10). Analyses of Cu, Zn, and

Pb were carried out only for 11 samples (Table 4), due to limited sample quantity, and not shown in Fig. 10. Their enrichment factors varied greatly, between 1 and 100.

## 5 Discussion

### 5.1 Source of Asian dust

Previous backward trajectory analysis (Jeong, 2008) and satellite remote sensing data (Husar et al., 2001; Zhang et al., 2003; Xuan et al., 2004; Seinfeld et al., 2004; McKendry et al., 2008; Jeong, et al., 2014) indicated that Mongolian and northern Chinese deserts are the source of Asian dust transported to Korea. The satellite images of

dust in this study confirmed that most dust outbreaks occurred in the Gobi Desert and sandy deserts distributed from southern Mongolia to northern China (Fig. 1a). The Taklamacan Desert is another source of Asian dust west of the Gobi Desert (Zhang, et al., 2003; Xuan et al., 2004). Unfortunately, the Taklamakan Desert, was not included in the available satellite images. Although Fig. 1a and Supplementary Fig. 1 are based on only one satellite image for the outbreak of each dust event, serial images acquired at an interval of 1 h or below around dust outbreaks

revealed no notable migration of dust storms from the west of the Gobi Desert. Mineral compositions of Asian dust coincided with fine (<20 µm) surface soils sampled in the Mongolian Gobi Desert (Fig. 5). Lower calcite and higher gypsum contents are attributed to atmospheric reactions. Major element compositions of Asian dust also coincided with those of Mongolian Gobi Desert soils (Fig. 6).

### 5.2 Path dependence of Asian dust geochemistry

Large temporal variation of Ca contents is a prominent feature of Asian dust, in contrast to the small variations in





Fe, Ti, K, Al, and Si contents (Fig. 9). Mg and Na contents also varied significantly, and showed positive correlations with Ca content (Fig. 9, Supplementary Fig. S3). Migration path data of the individual dust event in Supplementary Fig. S1 showed that Ca content is associated with the migration path of Asian dust. In total, 40% of the Asian dust storms migrating to the Korean Peninsula crossed the Chinese Loess Plateau (D3–6, D7, D10–11, D12, D19–20, and D21–23 in Fig. 9). All of the dust storms passing over the Loess Plateau were enriched with Ca (enrichment factor > 2.0 in Fig. 9).

Potential origins of the high-Ca dust storm are proposed here. The first is the entrainment of dust particles from the Loess Plateau. Although most dust storms originate from the deserts northwest of Loess Plateau, the storms may continuously entrain fine particles from the Loess Plateau in their early stages, i.e., just after leaving the desert. Loess is a loose eolian sediment comprising fine silt particles, and is probably susceptible to wind erosion. The average UCC-normalized Ca content of source soil samples (Table S2), excluding three ouliers (G2, G19, and G28), was 2.2 versus 2.7–2.9 for loess (Jeong et al., 2008, 2011). Abundant pedogenic calcite was derived via the dissolution of primary calcite in climates wetter than desert (Jeong et al., 2008, 2011). In addition, the satellite image showed a dust outbreak over Loess Plateau in 2013 (D13) (Supplementary Fig. S1). However, the mass emission of dust particles from the Loess Plateau is contradictory to current understanding. Most observation and dust emission modeling data showed that the Chinese Loess Plateau is a major sink for, but a very minor source of, dust (Zhang et al., 2003; Xuan et al., 2004).

The second potential origin of the high-Ca dust storm is the sandy deserts of northern China, which are distributed between the Mongolian Gobi Desert and the Loess Plateau. The mineralogy and geochemistry of fine soil fractions of the Mongolian Gobi Desert (Supplementary Tables S1–3) and the Loess Plateau (Supplementary Table S4) have been well characterized in this study and previous work (Jahn et al., 2001). However, the fine soil fractions of the sandy deserts lying between the Gobi Desert and the Loess Plateau were not investigated in this study and have rarely been addressed in previous works. The Gobi Desert ranges from southern Mongolia to northern China and is covered with silt, sand, gravel, rocky outcrops, and sparse vegetation with scattered dunes; in contrast, the sandy deserts in northern China (Badain Jaran, Tengger, Wulanbuhe, Kubuqi, and Maowusu deserts) are covered with dune fields. Subsaline to hypersaline and dry lakes are particularly common in the interdune basin of the Badain Jaran and Tengger deserts (Yang et al., 2003; Yang et al., 2011). Na, Ca, and Mg are the major cations in the hypersaline lake waters. Calcareous cementation is common on the surfaces of paleodunes. Calcareous lacustrine sediments deposited when the lake level was high are distributed around lakes and dry basins (Yang et



al., 2003). Calcareous cements and deposits composed of soft carbonate minerals are vulnerable to sand blasting occurring during dust storms, which supplies calcareous dust to migrating storms. A fraction of Mg and Na may also be present in the form of carbonates. In this study, XRD analyses of the samples from dust events that passed over the Loess Plateau revealed a weak peak corresponding to natron ($Na_2CO_3 \cdot 10H_2O$). The role of the sandy desert of northern China in the major element composition of Asian dust merits further investigation.

In this stage of research, the origin of high-Ca dust storm cannot be fully resolved because analytical data are insufficient, particularly in the sandy deserts of northern China. The origin of high-Ca dust could be clarified by further mineralogical and geochemical investigation of the sandy deserts and the estimation on the possibility of dust emission from loess plateau.

**5.3 Fractionation of minerals in dust**

The mineral compositions of the dust samples were generally consistent with those of source soils, indicating that little mineralogical fractionation occurred during dust storm outbreak and migration. It is noted that the coarsest dust (D11) was enriched with quartz and feldspars but relatively depleted in clay minerals. The dust samples for
one individual event (set 6; see Fig. 4) showed systematic changes in mineral contents, i.e., increasing clay mineral content and decreasing quartz and feldspar contents, toward the end of the dust event. The temporal changes in the mineralogy of the set 6 samples (D21–23) appear to be attributable to a decrease in particle size. Size distribution curves in Fig. 2 show the gradual decrease of coarse fractions (> 10 μm) from D21 to D23. In samples of sets 1 and 5, temporal changes of mineral composition were not evident and difficult to explain. These may be related to the
meteorological conditions during storm outbreak, migration, gravity settling, mixing or cloud processing. Mineral compositions of dust collected at different sites during individual events (sample sets 2–4; see Fig. 4) showed no obvious differences among sites, indicating that Korean Peninsula is likely too narrow to exhibit spatial variation of Asian dust properties.

**5.4 Fractionation of trace elements between soils and dust**

The ranges of the major element contents of the Asian dust samples were consistent with those of source soils,





while trace elements in the dust were fractionated from soils, showing lower Y, Zr, Hf, Ta, and heavy REE contents (Figs 7 and 8). Preferential depletion of heavy REEs suggests depletion of zircon, which is a mineral known to host heavy REEs (Henderson, 1984). Gravity settling of trace heavy minerals, particularly zircon, during the migration may be responsible for the fractionation of trace elements between soil and dust.

   The chondrite-normalized $(La/Yb)_N$ ratio (chondrite values by Boynton (1984)) represents the fractionation
of heavy REEs from light REEs. The average $(La/Yb)_N$ ratio of Asian dust in this study was 11.6, which was considerably higher than that of the source soil samples (8.7), probably due to the depletion of heavy REE-rich zircon. Meanwhile, the average europium (Eu) anomaly of dust (0.71) was not different from that of soil (0.70), which showed little fractionation (Table 4 and Supplementary Table S3) because Eu is hosted by plagioclase. The Chinese Loess Plateau is a sink of Asian dust neighboring on deserts. The $(La/Yb)_N$ ratio of Chinese loess are 8.7
in this study (average for 44 samples) (Supplementary Table S4) and 9.0 in Jahn et al. (2001; average for 30 samples) (Supplementary Table S5). The Eu anomaly for the Chinese loess showed little fractionation (0.65 in this study, and 0.64 in Jahn et al., 2001) from soils (0.70). These data indicate that Asian dust deposited on the Loess Plateau neighboring on the dust source experienced little fractionation.

   Analytical data of REE compositions are rare to find in previous works on Asian dust transported over long
distances. Lee et al. (2010) measured REEs in Asian dust sampled at three sites, derived from a dust event that took place in Korea during the period April 24–25, 2006. The average $(La/Yb)_N$ and Eu/Eu* ratios recalculated based on from their REE data were 11.5 and 0.55, respectively (Supplementary Table S6). The $(La/Yb)_N$ ratio obtained by Lee et al. (2010) is consistent with the values obtained in this study, although the Eu anomaly is somewhat low. The REE contents of Asian dust originated from the Gobi Desert and transported to the St. Elias Mountains (Yukon
Territory, Canada) were reported by Zdanowicz et al. (2007). The $(La/Yb)_N$ and Eu/Eu* ratios calculated based on the six REE data in the Yukon dust (excluding local dust) are 11.1 and 0.65, respectively (Supplementary Table S6), which are consistent with those for Korean dust. These findings support that the geochemical and mineralogical characteristics of Asian dust sampled in the Yukon Territory were little different from the dust sampled in Korea. Remarkably, the modal volume diameters of Asian dust particles are uniform among samples collected at the
western margin of the North Pacific (this study), in North Pacific Ocean sediments (Serno et al., 2014), and in the subarctic mountains of the North America (Zdanowicz et al., 2007). Previous studies showed that dust did not show significant changes in size distribution beyond transport distance of ~2,000 km, (Nakai et al., 1993; Rea,1994; Rea and Hovan,1995; Serno et al., 2014). Rather uniform properties of trans-Pacific Asian dust indicate that REE


fractionation of Asian dust occurred in a distance of ~ 2000 km from sources.

Deep-sea sediment samples from the central North Pacific Ocean were investigated in terms of REE contents to elucidate paleoclimatic changes. Since the sediments are normally mixtures of local volcanogenic particles and long-range transport eolian dust from Asia, the Chinese loess (and related sediments) are usually selected as an endmember of eolian component to estimate the accumulation rate and provenance of eolian particles. The

$(La/Yb)_N$ ratios of Asian dust fractions separated from pelagic sediments from the North Pacific were recorded as 9.1 (three central North Pacific samples, Nakai et al., 1993), 7.3 (11 samples from southern transect, Serno et al., 2014), and 7.7 (type 1 samples, Hyeong et al., 2004) (Supplementary Tables S7). These values are much lower than those of the Asian dust transported long distances in Zdanowicz et al. (2007), Lee et al. (2010), and this study. The Eu anomaly of pelagic sediments (0.66, Nakai et al. 2004; 0.74, Hyeong et al. 2004; 0.73, Serno et al. 2014)

(Supplementary Tables S7) is similar to that of Asian dust (0.71, this study; 0.65, Zdanowicz et al., 2007). REE data on long-range transport Asian dust in this study may improve the usefulness of REE as a proxy of paleoclimatic change.

## 5.5 Mixing and reaction of Asian dust with polluted air

Asian dust passes through industrialized regions of East Asia, where air pollution is a severe environmental issue. Asian dust mixes and reacts with polluted air (Nishikawa et al., 1991; McKendry, et al., 2008; Huang et al., 2010). The concentration of atmospheric pollutants is high particularly in winter and spring seasons which are also the major seasons of dust outbreaks. Calcite, which is the most reactive mineral in dust, reacts with acidic gases in the

atmosphere (mostly sulfur species) originating from pollution (Dentener et al., 1996; Laskin et al., 2005; Matsuki et al., 2005; Jeong and Chun, 2006). Higher gypsum and lower calcite contents of Asian dust compared to source soils (Table 2, Supplementary Table S1, and Fig. 5) suggest the conversion of calcite into gypsum during the transport. This conversion is also supported by the similar average Ca contents of both the dust (5.1%) and soils (6.4%). During one of the individual dust events in this study (set 6 samples; see Fig. 4), the increased gypsum

content and concomitantly decreased calcite content were clearly the result of progressive reaction of dust with air pollutants toward the end of the dust event. During the dust event of set 3, the gypsum content of dust was greatly enhanced concomitantly with the decrease of calcite content at Deokjeok Island (D14), while it was lower than the calcite content at Andong (D13). This supports the conversion of calcite to gypsum in the severely polluted



atmosphere around the densely populated metropolitan region of Seoul, Korea. Marked variation of both the calcite (0.5~11%) and gypsum (0~18%) contents of dust is intuitive because pollutant concentrations vary widely with regional weather conditions.

In Asian dust, Cu, Zn, Sn, and Pb (Fig. 7) are the major heavy metal pollutants of combustion origin (Duan and Tan, 2013). Although the contents of Cu, Zn, and Pb were not analyzed in all of the samples due to small dust

quantities, the marked temporal variation of Sn content represent the temporal variation of heavy metal pollutants (Fig. 10). The intra-event dust samples of set 6 showed a gradual increase of Sn content toward the end of the dust event, while the contents of REEs and Zr of soil origin decreased progressively, consistent with progressive mixing of dust with regional pollutants (Fig. 10). The Sn contents were low in samples of sets 2 and 4, showing only minor variations, but it was high in D14 (set 3) at Deokjeok Island, consistent with high gypsum content.

## 6 Summary and Conclusions

Systematic analyses of Asian dust samples collected over a long period showed the mineralogical and geochemical properties consistent with those of fine silt fractions (< 20 μm) of the source soils. Clay minerals were most

abundant, followed by quartz, plagioclase, K-feldspar, calcite, and gypsum. Asian dust crossing the Loess Plateau has a higher Ca content, entraining calcite-rich fine dust particles that probably originated from calcareous sediments in northern China sandy deserts and pedogenic calcite-rich loess. Dust-laden air parcels mix with atmospheric pollutants over East Asia. Calcite was found to react with pollutants to form gypsum. Serial dust samples for each dust event show scant changes in major-element contents. However, trace element contents varied

widely due to REE fractionation and mixing with polluted air enriched with heavy metals of pollution origin. Selective depletion of heavy REEs in dust from source soils resulted in increased $(La/Yb)_N$ ratios. The samples of one dust event showed a trend of increasing clay minerals and gypsum contents toward the end of the event in association with decreasing particle size and progressive reactions. Mineral dust transported over long distances is the subject of much interdisciplinary research. This study describes not only the average properties of dust, but also

inter-event variations therein and fractionation from source soils, thereby providing a basis for climatic and atmospheric reaction modeling, and for analysis of deep-sea sediments, fine-grained soils, and ice-sheet dust.





## Acknowledgements

This study was funded by the National Research Foundation of Korea grant NRF-2017R1A2B2011422 to G.Y. Jeong.

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





Table 1. List of Asian dust sampled in Korea and desert soils from Mongolian Gobi desert.

| Asian dust | | | | | | | Mongolian Gobi desert soil | | |
|---|---|---|---|---|---|---|---|---|---|
| Sample | Site[1] | Year | Month/Date (sampling hour) | Travel (h)[2] | Dist. (km)[3] | Conc.[4] (μg/m³) | Sample | Latitude | Longitude |
| D1 | AD | 2005 | 4/20(09-17) | - | - | 366 | G1 | N 45º 19' 21.39" | E 106º 32' 47.40" |
| D2 | AD | 2008 | 5/31(08-18) | 54 | 1800 | 292 | G2 | N 44º 40' 39.67" | E 106º 56' 13.41" |
| D3 | AD | 2009 | 3/16(09-18) | | | | G3 | N 44º 26' 46.20" | E 107º 08' 21.72" |
| D4 | AD | 2009 | 3/16(18)-17(09) | 45 | 1900 | 428 | G4 | N 44º 14' 17.02" | E 107º 32' 07.85" |
| D5 | AD | 2009 | 3/17(09-18) | | | | G5 | N 44º 04' 00.11" | E 107º 43' 32.88" |
| D6 | AD | 2009 | 3/17(19)-18(09) | | | | G6 | N 44º 04' 01.97" | E 108º 16' 22.73" |
| D7 | AD | 2010 | 3/20(19)-21(10) | 30 | 2200 | 1788 | G7 | N 43º 37' 48.48" | E 108º 33' 42.42" |
| D8 | AD | 2011 | 5/1(09-18) | 41 | 1300 | 300 | G8 | N 43º 27' 28.00" | E 109º 01' 39.26" |
| D9 | AD | 2011 | 5/12(20)-13(11) | 49 | 1700 | 322 | G9 | N 43º 14' 50.26" | E 108º 10' 02.68" |
| D10 | SL | 2012 | 3/31(09)-4/1(08) | 25 | 1700 | 215 | G10 | N 43º 11' 28.31" | E 107º 01' 11.78" |
| D11 | DJ | 2012 | 3/31(09)-4/1(08) | | | 220 | G11 | N 42º 38' 16.75" | E 107º 18' 45.96" |
| D12 | SL | 2013 | 3/9(09)-10(08) | - | 1500 | 215 | G12 | N 42º 32' 24.88" | E 106º 55' 27.65" |
| D13 | AD | 2014 | 3/18(10-22) | 42 | 1800 | 378 | G13 | N 42º 32' 41.71" | E 106º 29' 27.33" |
| D14 | DJ | 2014 | 3/18(09-24) | | | 214 | G14 | N 42º 28' 43.44" | E 106º 02' 20.82" |
| D15 | SL | 2015 | 2/22(09)-23(08) | | | 1044 | G15 | N 42º 43' 57.89" | E 105º 28' 30.02" |
| D16 | AD | 2015 | 2/22(16)-23(18) | 23 | 1400 | 469 | G16 | N 43º 06' 27.97" | E 104º 58' 05.30" |
| D17 | DJ | 2015 | 2/22(09)-23(08) | | | 1037 | G17 | N 43º 05' 29.85" | E 104º 13' 29.36" |
| D18 | AD | 2015 | 3/1(17)-2(11) | 29 | 1600 | 281 | G18 | N 42º 51' 36.78" | E 104º 08' 24.24" |
| D19 | SL | 2016 | 3/6(09)-7(08) | 44 | 1700 | 168 | G19 | N 42º 31' 21.33" | E 103º 51' 15.57" |
| D20 | SL | 2016 | 3/7(09)-8(08) | | | | G20 | N 42º 43' 41.69" | E 103º 45' 58.64" |
| D21 | AD | 2017 | 5/6(13-21) | | | | G21 | N 42º 52' 35.91" | E 103º 46' 44.04" |
| D22 | AD | 2017 | 5/6(21)-7(19) | 60 | 1700 | 331 | G22 | N43º 10' 52.04" | E 104º 00' 38.56" |
| D23 | AD | 2017 | 5/8(14)-9(10) | | | | G23 | N 43º 21' 00.20" | E 103º 34' 18.01" |
| D24 | AD | 2018 | 4/6(15)-7(09) | 47 | 2400 | 194 | G24 | N 43º 10' 55.20" | E 103º 07' 34.28" |
| D25 | AD | 2018 | 4/15(15)-16(10) | 43 | 1300 | 316 | G25 | N 42º 56' 04.51" | E 102º 01' 39.43" |
| | | | | | | | G26 | N 42º 56' 43.06" | E 101º 39' 04.13" |
| | | | | | | | G27 | N 43º 06' 51.96" | E 101º 04' 28.67" |
| | | | | | | | G28 | N 43º 18' 38.66" | E 101º 05' 31.79" |
| | | | | | | | G29 | N 43º 26' 55.40" | E 101º 15' 28.02" |
| | | | | | | | G30 | N 43º 56' 49.43" | E 101º 27' 19.13" |
| | | | | | | | G31 | N 44º 02' 09.93" | E 101º 30' 26.83" |
| | | | | | | | G32 | N 44º 25' 27.10" | E 101º 33' 49.81" |
| | | | | | | | G33 | N 44º 41' 36.96" | E 101º 52' 59.77" |
| | | | | | | | G34 | N 45º 05' 25.65" | E 102º 18' 03.50" |

[1] Dust sampling sites
AD: Andong, 36º 32' 34.76", 128º 47' 54.92"
DJ: Deokjeokdo, 37º 13' 59.47", 126º 08' 56.70"
SL: Seoul, 37º 36' 05.20", 127º 02' 49.02"
[2] Travel time when $PM_{10}$ concentration abruptly increased in the monitoring stations in Korea.
[3] Distance from the eastern boundary of dust air parcel in outbreak area to Seoul, Korea.
[4] Peak concentration of $PM_{10}$ measured in the nearby monitoring stations operated by Korea Meteorological Administration.





Table 2. Mineral compositions of Asian dust (wt.%) determined by X-ray diffraction. Samples indicated by shades were collected during the same dust event.

| Sample | Q[1] | P | Kf | ISCMs | Ch | Ka | A | Ca | D | G | H | Total clays | P/kf |
|---|---|---|---|---|---|---|---|---|---|---|---|---|---|
| D1 | 23.6 | 14.6 | 5.4 | 38.4 | 4.6 | 2.6 | 2.2 | 4.4 | 1.2 | 2.7 | 0.0 | 45.7 | 2.7 |
| D2 | 24.2 | 13.9 | 6.8 | 41.6 | 3.1 | 1.4 | 2.7 | 3.1 | 2.0 | 1.3 | 0.0 | 46.2 | 2.1 |
| D3 | 23.5 | 14.3 | 3.8 | 35.1 | 3.4 | 2.1 | 1.0 | 5.4 | 1.9 | 9.5 | 0.0 | 40.6 | 3.7 |
| D4 | 18.0 | 12.2 | 4.7 | 35.1 | 5.9 | 3.1 | 2.4 | 4.4 | 2.4 | 11.7 | 0.0 | 44.0 | 2.6 |
| D5 | 21.1 | 12.7 | 4.8 | 33.8 | 5.4 | 2.8 | 1.7 | 6.8 | 2.6 | 8.3 | 0.0 | 42.0 | 2.7 |
| D6 | 18.2 | 10.9 | 4.8 | 34.1 | 5.7 | 3.1 | 2.1 | 8.7 | 2.6 | 9.7 | 0.0 | 42.9 | 2.3 |
| D7 | 21.5 | 13.0 | 3.4 | 42.4 | 4.4 | 3.1 | 0.8 | 6.3 | 1.0 | 4.2 | 0.0 | 49.9 | 3.8 |
| D8 | 18.7 | 12.6 | 6.0 | 45.9 | 4.5 | 2.1 | 2.7 | 3.8 | 1.1 | 2.5 | 0.0 | 52.5 | 2.1 |
| D9 | 21.9 | 14.0 | 5.7 | 43.9 | 4.1 | 1.1 | 1.9 | 4.6 | 2.5 | 0.2 | 0.0 | 49.1 | 2.5 |
| D10 | 28.7 | 17.5 | 5.6 | 33.5 | 3.5 | 2.3 | 0.9 | 6.3 | 0.8 | 0.8 | 0.0 | 39.3 | 3.1 |
| D11 | 31.0 | 17.5 | 6.8 | 28.5 | 3.8 | 1.5 | 2.5 | 6.2 | 1.4 | 0.5 | 0.0 | 33.8 | 2.6 |
| D12 | 19.1 | 12.2 | 7.1 | 33.7 | 2.2 | 1.8 | 2.1 | 1.2 | 2.4 | 18.3 | 0.0 | 37.8 | 1.7 |
| D13 | 17.9 | 12.3 | 6.1 | 47.5 | 3.8 | 1.3 | 1.9 | 5.5 | 0.6 | 3.0 | 0.0 | 52.6 | 2.0 |
| D14 | 18.3 | 13.4 | 5.3 | 47.9 | 1.8 | 1.9 | 1.0 | 0.5 | 0.4 | 9.4 | 0.0 | 51.6 | 2.5 |
| D15 | 20.1 | 10.9 | 3.1 | 53.2 | 3.7 | 2.2 | 1.0 | 4.1 | 0.7 | 1.0 | 0.0 | 59.1 | 3.5 |
| D16 | 19.7 | 11.3 | 3.7 | 49.5 | 3.7 | 2.7 | 1.0 | 3.4 | 0.6 | 4.3 | 0.0 | 56.0 | 3.0 |
| D17 | 19.7 | 10.4 | 4.2 | 51.5 | 3.7 | 2.6 | 0.7 | 3.8 | 0.3 | 2.1 | 0.9 | 57.9 | 2.5 |
| D18 | 19.0 | 10.5 | 4.3 | 51.9 | 3.1 | 1.7 | 1.3 | 2.8 | 0.5 | 5.1 | 0.0 | 56.7 | 2.4 |
| D19 | 17.4 | 9.9 | 4.5 | 47.4 | 2.8 | 2.7 | 1.2 | 6.2 | 1.5 | 6.4 | 0.0 | 53.0 | 2.2 |
| D20 | 17.1 | 10.5 | 5.7 | 39.2 | 4.1 | 2.0 | 2.4 | 9.2 | 1.9 | 8.0 | 0.0 | 45.3 | 1.9 |
| D21 | 24.7 | 14.9 | 5.0 | 33.9 | 3.9 | 2.0 | 3.0 | 11.0 | 1.4 | 0.0 | 0.0 | 39.9 | 3.0 |
| D22 | 19.9 | 10.5 | 4.0 | 41.3 | 4.4 | 2.0 | 3.8 | 8.5 | 2.4 | 3.3 | 0.0 | 47.7 | 2.7 |
| D23 | 14.1 | 9.0 | 4.9 | 46.4 | 5.0 | 2.4 | 2.6 | 5.2 | 2.6 | 7.9 | 0.0 | 53.8 | 1.9 |
| D24 | 19.0 | 10.6 | 6.1 | 46.4 | 4.8 | 1.3 | 3.0 | 3.8 | 1.3 | 3.6 | 0.0 | 52.5 | 1.8 |
| D25 | 19.6 | 11.4 | 6.0 | 49.5 | 4.4 | 1.2 | 2.2 | 2.9 | 0.8 | 2.1 | 0.0 | 55.1 | 1.9 |
| Average | 20.6 | 12.4 | 5.1 | 42.1 | 4.0 | 2.1 | 1.9 | 5.1 | 1.5 | 5.0 | 0.0 | 48.2 | 2.5 |
| St.dev. | 3.7 | 2.2 | 1.1 | 7.1 | 1.0 | 0.6 | 0.8 | 2.5 | 0.8 | 4.4 | | 6.9 | 0.6 |
| Mineral compositions determined by SEM single particle analysis | | | | | | | | | | | | | |
| D13 | 19.0 | 10.9 | 3.8 | 53.8 | 2.4 | 1.1 | 0.7 | 7.0 | 0.8 | 0.5 | 0.0 | 57.3 | 2.9 |
| D18 | 19.5 | 9.3 | 3.4 | 55.7 | 2.6 | 2.3 | 0.3 | 4.4 | 0.2 | 2.1 | 0.0 | 60.7 | 2.7 |
| D19 | 16.9 | 7.9 | 2.6 | 50.2 | 4.9 | 2.3 | 0.6 | 9.4 | 2.5 | 2.7 | 0.0 | 57.4 | 3.1 |
| D22 | 19.0 | 11.7 | 3.7 | 43.2 | 6.3 | 3.5 | 0.2 | 8.8 | 0.7 | 2.8 | 0.0 | 53.0 | 3.1 |
| D24 | 21.0 | 9.7 | 4.4 | 50.9 | 4.0 | 1.5 | 0.8 | 4.8 | 0.8 | 2.0 | 0.0 | 56.4 | 2.2 |
| SEM[2] | 19.1 | 9.9 | 3.6 | 50.8 | 4.1 | 2.1 | 0.5 | 6.9 | 1.0 | 2.0 | 0.0 | 56.9 | 2.8 |
| XRD[3] | 18.6 | 10.8 | 5.0 | 46.9 | 3.8 | 1.8 | 2.2 | 5.4 | 1.3 | 4.3 | 0.0 | 52.5 | 2.2 |

[1]Q=quartz, P=plagioclase, Kf=K-feldspar, ISCMs=illite-smectite series clay minerals, Ch=chlorite, Ka=kaolinite, A=amphibole, Ca=calcite, D=dolomite, G=gypsum, H=halite. [2] [3]Average of five samples.





Table 3. Major element composition of Asian dust (unit in wt.%) on volatile-free basis. Samples indicated by shade were collected at different sites or serially during the dust event.

| Sample | Si | Al | Fe | Mn | Mg | Ca | Na | K | Ti | P | Total |
|---|---|---|---|---|---|---|---|---|---|---|---|
| D1 | 28.08 | 9.32 | 5.27 | 0.11 | 2.00 | 3.97 | 1.05 | 2.68 | 0.52 | 0.12 | 53.11 |
| D2 | 28.48 | 9.17 | 5.19 | 0.12 | 1.83 | 3.54 | 1.20 | 2.75 | 0.53 | 0.16 | 52.97 |
| D3 | 26.65 | 8.61 | 4.83 | 0.09 | 2.40 | 6.62 | 1.78 | 2.49 | 0.48 | 0.11 | 54.06 |
| D4 | 25.67 | 9.17 | 5.28 | 0.11 | 2.65 | 6.59 | 1.57 | 2.71 | 0.48 | 0.12 | 54.34 |
| D5 | 25.66 | 9.62 | 5.16 | 0.10 | 2.46 | 6.68 | 1.39 | 2.54 | 0.49 | 0.10 | 54.22 |
| D6 | 24.98 | 9.08 | 5.17 | 0.11 | 2.68 | 7.93 | 1.47 | 2.77 | 0.47 | 0.12 | 54.75 |
| D7 | 26.93 | 9.24 | 5.15 | 0.11 | 2.39 | 5.30 | 1.26 | 2.69 | 0.51 | 0.11 | 53.69 |
| D8 | 27.24 | 9.69 | 5.54 | 0.11 | 2.20 | 3.87 | 1.17 | 2.80 | 0.54 | 0.19 | 53.35 |
| D9 | 28.11 | 9.02 | 5.35 | 0.11 | 2.05 | 3.82 | 1.05 | 2.83 | 0.51 | 0.26 | 53.12 |
| D10 | 27.72 | 8.38 | 4.95 | 0.11 | 2.15 | 5.34 | 1.91 | 2.42 | 0.51 | 0.12 | 53.61 |
| D11 | 28.82 | 8.47 | 4.69 | 0.10 | 1.76 | 4.61 | 1.50 | 2.47 | 0.51 | 0.12 | 53.06 |
| D12 | 25.19 | 9.00 | 5.56 | 0.15 | 1.99 | 7.27 | 1.89 | 3.03 | 0.53 | 0.19 | 54.78 |
| D13 | 26.64 | 9.51 | 5.23 | 0.14 | 2.31 | 4.46 | 1.93 | 2.91 | 0.53 | 0.16 | 53.80 |
| D14 | 27.59 | 9.37 | 5.35 | 0.13 | 1.89 | 3.80 | 1.42 | 3.16 | 0.55 | 0.16 | 53.42 |
| D15 | 27.85 | 9.73 | 5.26 | 0.13 | 2.08 | 3.58 | 1.01 | 2.80 | 0.54 | 0.12 | 53.09 |
| D16 | 27.59 | 9.60 | 5.38 | 0.13 | 2.10 | 3.91 | 1.09 | 2.80 | 0.53 | 0.13 | 53.26 |
| D17 | 27.62 | 9.58 | 5.29 | 0.13 | 2.10 | 3.63 | 1.53 | 2.78 | 0.50 | 0.12 | 53.28 |
| D18 | 28.09 | 9.44 | 5.17 | 0.12 | 2.03 | 3.61 | 1.21 | 2.79 | 0.52 | 0.11 | 53.09 |
| D19 | 25.73 | 9.25 | 5.35 | 0.12 | 2.67 | 5.93 | 1.69 | 2.90 | 0.51 | 0.14 | 54.29 |
| D20 | 25.22 | 8.89 | 5.29 | 0.12 | 2.53 | 7.45 | 1.71 | 2.86 | 0.51 | 0.15 | 54.71 |
| D21 | 27.77 | 8.55 | 4.93 | 0.12 | 1.98 | 5.80 | 1.15 | 2.62 | 0.49 | 0.14 | 53.54 |
| D22 | 25.96 | 9.36 | 5.76 | 0.12 | 2.47 | 5.98 | 1.11 | 2.66 | 0.50 | 0.17 | 54.09 |
| D23 | 25.11 | 9.64 | 5.77 | 0.12 | 3.12 | 6.02 | 1.07 | 2.91 | 0.50 | 0.14 | 54.38 |
| D24 | 27.75 | 9.62 | 5.31 | 0.11 | 2.11 | 3.68 | 1.09 | 2.90 | 0.50 | 0.12 | 53.19 |
| D25 | 28.36 | 9.78 | 5.43 | 0.115 | 1.90 | 2.80 | 0.84 | 2.82 | 0.55 | 0.19 | 52.78 |
| Average | 26.99 | 9.24 | 5.27 | 0.12 | 2.23 | 5.05 | 1.36 | 2.76 | 0.51 | 0.14 | 53.68 |
| St.dev. | 1.21 | 0.41 | 0.25 | 0.01 | 0.33 | 1.48 | 0.32 | 0.17 | 0.02 | 0.04 | 0.62 |





Table 4. Trace element composition of Asian dust (unit in ppm). Samples indicated by shade were collected at different sites or serially during the dust event.

| Sample | S | Sc | V | Cr | Co | Ni | Cu | Zn | Ga | Rb | Sr | Y | Zr |
|---|---|---|---|---|---|---|---|---|---|---|---|---|---|
| D.L.* | 10 | 1 | 5 | 20 | 1 | 1 | 1 | 1 | 1 | 1 | 2 | 0.5 | 1 |
| D1 | - | 14 | 113 | 110 | 22 | - | - | - | 20 | 111 | 225 | 27.6 | 128 |
| D2 | - | 13 | 113 | 110 | 33 | - | - | - | 18 | 107 | 217 | 26.9 | 139 |
| D3 | - | 12 | 97 | 100 | 16 | - | - | - | 17 | 91 | 297 | 23.6 | 128 |
| D4 | 2190 | 14 | 116 | 100 | 19 | 55 | 1280 | 500 | 19 | 101 | 326 | 24.9 | 119 |
| D5 | - | 12 | 100 | 100 | 20 | - | - | - | 17 | 82 | 268 | 21.3 | 106 |
| D6 | - | 13 | 111 | 190 | 17 | - | - | - | 18 | 99 | 315 | 23.2 | 108 |
| D7 | 899 | 15 | 114 | 80 | 19 | 53 | 155 | 239 | 20 | 103 | 275 | 28.4 | 137 |
| D8 | - | 13 | 101 | 100 | 19 | - | - | - | 19 | 101 | 208 | 25.7 | 111 |
| D9 | - | 14 | 107 | 100 | 18 | - | - | - | 18 | 112 | 210 | 26.4 | 129 |
| D10 | 1150 | 12 | 99 | 100 | 19 | 65 | 448 | 823 | 15 | 86 | 251 | 24.4 | 136 |
| D11 | 440 | 12 | 96 | 100 | 17 | 54 | 218 | 415 | 18 | 103 | 237 | 26.7 | 148 |
| D12 | - | 10 | 141 | 160 | 17 | - | - | - | 20 | 91 | 242 | 19.8 | 144 |
| D13 | - | 12 | 102 | 90 | 18 | - | - | - | 18 | 91 | 224 | 22.4 | 106 |
| D14 | 2500 | 12 | 113 | 100 | 18 | 68 | 1330 | 900 | 23 | 109 | 218 | 26.5 | 141 |
| D15 | 420 | 15 | 118 | 190 | 20 | 53 | 166 | 157 | 21 | 115 | 218 | 29.7 | 125 |
| D16 | 983 | 15 | 119 | 120 | 20 | 56 | 401 | 236 | 20 | 110 | 219 | 28.0 | 121 |
| D17 | 581 | 14 | 104 | 80 | 19 | 51 | 219 | 184 | 20 | 108 | 203 | 26.6 | 108 |
| D18 | 1190 | 13 | 108 | 90 | 19 | 53 | 2840 | 349 | 20 | 108 | 217 | 27.7 | 121 |
| D19 | - | 13 | 97 | 110 | 18 | 34 | 1436 | 733 | 19 | 99 | 298 | 24.9 | 141 |
| D20 | - | 12 | 95 | 100 | 16 | - | - | - | 17 | 88 | 290 | 21.8 | 144 |
| D21 | - | 13 | 99 | 140 | 19 | - | - | - | 17 | 96 | 249 | 25.2 | 121 |
| D22 | - | 14 | 123 | 110 | 21 | - | - | - | 19 | 100 | 272 | 26.6 | 116 |
| D23 | - | 14 | 116 | 110 | 19 | - | - | - | 20 | 102 | 309 | 23.8 | 109 |
| D24 | - | 14 | 105 | 100 | 19 | - | - | - | 19 | 109 | 210 | 25.7 | 109 |
| D25 | 673 | 15 | 117 | 80 | 20 | 50 | 692 | 378 | 20 | 115 | 188 | 28.8 | 134 |
| Average | 1103 | 13 | 109 | 111 | 19 | 54 | 835 | 447 | 19 | 101 | 247 | 25.5 | 125 |

*Detection limit (ppm)





| Sample | Nb | Sn | Sb | Cs | Ba | Hf | Ta | Tl | Pb | Th | U | La | Ce |
|---|---|---|---|---|---|---|---|---|---|---|---|---|---|
| D.L.* | 0.2 | 1 | 0.2 | 0.1 | 2 | 0.1 | 0.01 | 0.05 | 3 | 0.05 | 0.01 | 0.05 | 0.05 |
| D1 | 14.3 | 12 | 1.3 | 8.6 | 620 | 4.0 | 0.82 | 0.89 | - | 12.7 | 2.72 | 46.7 | 90.7 |
| D2 | 12.4 | 61 | 0.3 | 8.6 | 622 | 3.7 | 0.72 | 0.72 | - | 12.2 | 2.73 | 41.3 | 81.0 |
| D3 | 10.6 | 26 | 0.9 | 7.9 | 549 | 3.9 | 0.73 | 0.62 | - | 11.2 | 3.59 | 36.0 | 70.6 |
| D4 | 11.6 | 29 | 4.5 | 9.1 | 571 | 3.4 | 0.69 | 0.86 | 250 | 11.9 | 3.78 | 35.5 | 71.7 |
| D5 | 10.3 | 40 | 1.3 | 7.4 | 546 | 3.1 | 0.34 | 0.68 | - | 9.8 | 3.04 | 30.7 | 61.6 |
| D6 | 11.0 | 39 | 2.6 | 8.4 | 589 | 3.0 | 0.69 | 0.60 | - | 11.3 | 3.64 | 33.1 | 66.5 |
| D7 | 11.8 | 4 | 0.7 | 8.6 | 562 | 3.6 | 0.73 | 0.50 | 66 | 12.1 | 3.24 | 41.1 | 82.4 |
| D8 | 12.1 | 7 | < 0.2 | 8.3 | 519 | 3.8 | 0.65 | 0.56 | - | 11.2 | 2.84 | 37.3 | 75.7 |
| D9 | 11.6 | 6 | < 0.2 | 9.0 | 568 | 4.2 | 0.69 | 0.68 | - | 11.8 | 2.69 | 36.6 | 73.2 |
| D10 | 13.0 | 5 | 0.2 | 6.5 | 589 | 3.9 | 0.59 | 0.44 | 199 | 10.9 | 2.50 | 44.1 | 85.5 |
| D11 | 11.6 | 4 | 0.8 | 7.0 | 650 | 4.4 | 0.86 | 0.48 | 80 | 11.9 | 2.79 | 45.3 | 87.9 |
| D12 | 10.8 | 96 | 4.0 | 7.7 | 788 | 4.0 | 0.67 | 1.17 | - | 10.1 | 2.85 | 34.4 | 63.5 |
| D13 | 10.3 | 10 | 0.9 | 7.6 | 507 | 2.9 | 0.55 | 0.61 | - | 10.0 | 2.73 | 33.0 | 65.4 |
| D14 | 11.5 | 41 | 2.9 | 8.9 | 699 | 4.2 | 0.77 | 0.87 | 264 | 12.6 | 3.07 | 38.6 | 77.1 |
| D15 | 11.9 | 3 | 0.5 | 9.3 | 606 | 3.7 | 0.89 | 0.50 | 41 | 12.9 | 2.72 | 41.7 | 84.9 |
| D16 | 12.5 | 3 | < 0.2 | 9.0 | 612 | 3.6 | 0.77 | 0.47 | 78 | 12.2 | 2.71 | 40.5 | 82.7 |
| D17 | 11.3 | 2 | < 0.2 | 8.7 | 569 | 3.3 | 0.66 | 0.27 | 61 | 11.5 | 2.35 | 37.5 | 74.4 |
| D18 | 12.7 | 3 | < 0.2 | 8.8 | 599 | 3.5 | 0.79 | 0.53 | 104 | 11.7 | 2.59 | 38.0 | 77.8 |
| D19 | 12.2 | 56 | 5.0 | 8.0 | 747 | 3.9 | 0.77 | 0.48 | 72.5 | 11.7 | 3.43 | 35.9 | 72.3 |
| D20 | 10.4 | 39 | 1.9 | 6.7 | 778 | 4.1 | 0.58 | 0.52 | - | 10.9 | 3.28 | 32.6 | 64.4 |
| D21 | 13.0 | 6 | 1.0 | 6.8 | 627 | 3.1 | 0.75 | 0.31 | - | 11.0 | 2.85 | 44.7 | 89.2 |
| D22 | 12.1 | 7 | 2.4 | 8.2 | 625 | 3.5 | 0.84 | 0.55 | - | 12.2 | 3.30 | 43.8 | 84.4 |
| D23 | 10.6 | 14 | 8.5 | 9.3 | 583 | 3.3 | 0.66 | 0.97 | - | 12.3 | 3.96 | 37.1 | 71.5 |
| D24 | 12.4 | 3 | 1.1 | 8.9 | 597 | 3.2 | 0.79 | 0.53 | - | 11.5 | 2.39 | 38.6 | 77.5 |
| D25 | 13.9 | 5 | 1.5 | 9.5 | 602 | 4.0 | 0.93 | 0.50 | 195 | 13.5 | 2.72 | 42.7 | 88.2 |
| Average | 11.8 | 21 | 1.7 | 8.3 | 613 | 3.7 | 0.72 | 0.61 | 116 | 11.6 | 3.0 | 38.7 | 76.8 |

*Detection limit (ppm)





| Sample | Pr | Nd | Sm | Eu | Gd | Tb | Dy | Ho | Er | Tm | Yb | Lu | $(La/Yb)_N{}^{1)}$ | $Eu/Eu^{*1)}$ |
|---|---|---|---|---|---|---|---|---|---|---|---|---|---|---|
| D.L.* | 0.01 | 0.05 | 0.01 | 0.005 | 0.01 | 0.01 | 0.01 | 0.01 | 0.01 | 0.005 | 0.01 | 0.002 | | |
| D1 | 9.83 | 36.5 | 6.55 | 1.50 | 5.24 | 0.77 | 4.73 | 0.93 | 2.61 | 0.388 | 2.51 | 0.374 | 12.5 | 0.78 |
| D2 | 8.42 | 31.0 | 6.44 | 1.33 | 5.51 | 0.81 | 4.63 | 0.87 | 2.47 | 0.348 | 2.23 | 0.348 | 12.5 | 0.68 |
| D3 | 7.39 | 27.5 | 5.55 | 1.07 | 4.52 | 0.69 | 4.34 | 0.84 | 2.27 | 0.332 | 2.26 | 0.330 | 10.7 | 0.65 |
| D4 | 7.62 | 26.7 | 5.64 | 1.04 | 4.30 | 0.67 | 3.98 | 0.76 | 2.33 | 0.331 | 2.30 | 0.329 | 10.4 | 0.65 |
| D5 | 6.51 | 25.2 | 4.78 | 1.02 | 4.01 | 0.65 | 3.79 | 0.72 | 1.99 | 0.287 | 1.84 | 0.261 | 11.2 | 0.71 |
| D6 | 6.87 | 24.6 | 4.79 | 1.00 | 4.29 | 0.66 | 3.82 | 0.76 | 2.32 | 0.336 | 2.30 | 0.356 | 9.7 | 0.67 |
| D7 | 9.19 | 32.9 | 6.09 | 1.34 | 5.11 | 0.75 | 4.64 | 0.91 | 2.65 | 0.411 | 2.64 | 0.390 | 10.5 | 0.73 |
| D8 | 8.00 | 29.4 | 5.55 | 1.33 | 4.85 | 0.76 | 4.40 | 0.91 | 2.46 | 0.359 | 2.44 | 0.349 | 10.3 | 0.78 |
| D9 | 8.13 | 30.8 | 6.27 | 1.30 | 4.92 | 0.78 | 4.51 | 0.85 | 2.50 | 0.391 | 2.29 | 0.343 | 10.8 | 0.72 |
| D10 | 9.33 | 35.5 | 6.47 | 1.24 | 4.48 | 0.71 | 4.29 | 0.81 | 2.45 | 0.323 | 2.10 | 0.314 | 14.2 | 0.70 |
| D11 | 9.74 | 36.7 | 6.50 | 1.30 | 5.14 | 0.78 | 4.61 | 0.85 | 2.44 | 0.355 | 2.36 | 0.345 | 12.9 | 0.69 |
| D12 | 6.20 | 23.0 | 3.94 | 0.82 | 3.59 | 0.53 | 2.96 | 0.58 | 1.69 | 0.239 | 1.58 | 0.242 | 14.7 | 0.67 |
| D13 | 7.04 | 25.8 | 5.00 | 1.16 | 4.09 | 0.61 | 3.88 | 0.71 | 2.08 | 0.305 | 1.97 | 0.327 | 11.3 | 0.78 |
| D14 | 8.12 | 30.7 | 5.80 | 1.22 | 4.62 | 0.73 | 4.65 | 0.87 | 2.36 | 0.352 | 2.29 | 0.366 | 11.4 | 0.72 |
| D15 | 9.12 | 34.1 | 6.62 | 1.33 | 5.65 | 0.86 | 4.93 | 0.97 | 2.77 | 0.415 | 2.84 | 0.432 | 9.9 | 0.66 |
| D16 | 8.84 | 33.7 | 6.48 | 1.29 | 5.42 | 0.83 | 4.95 | 0.92 | 2.83 | 0.391 | 2.57 | 0.383 | 10.6 | 0.67 |
| D17 | 8.22 | 29.9 | 6.10 | 1.23 | 5.13 | 0.80 | 4.54 | 0.88 | 2.54 | 0.356 | 2.20 | 0.342 | 11.5 | 0.67 |
| D18 | 8.21 | 30.3 | 6.77 | 1.28 | 5.06 | 0.76 | 4.57 | 0.93 | 2.68 | 0.382 | 2.29 | 0.380 | 11.2 | 0.67 |
| D19 | 7.51 | 28.9 | 5.63 | 1.18 | 4.49 | 0.69 | 4.03 | 0.77 | 2.22 | 0.324 | 2.24 | 0.326 | 10.8 | 0.72 |
| D20 | 6.81 | 25.4 | 4.85 | 1.03 | 3.77 | 0.66 | 3.85 | 0.71 | 1.95 | 0.290 | 1.94 | 0.292 | 11.3 | 0.74 |
| D21 | 9.91 | 34.9 | 5.54 | 1.33 | 4.43 | 0.69 | 4.36 | 0.77 | 2.23 | 0.301 | 2.02 | 0.324 | 14.9 | 0.82 |
| D22 | 9.08 | 33.1 | 6.13 | 1.31 | 5.04 | 0.77 | 4.56 | 0.88 | 2.33 | 0.364 | 2.39 | 0.348 | 12.4 | 0.72 |
| D23 | 7.37 | 26.5 | 5.36 | 1.12 | 4.07 | 0.67 | 4.10 | 0.83 | 2.37 | 0.355 | 2.10 | 0.302 | 11.9 | 0.73 |
| D24 | 8.18 | 29.9 | 5.77 | 1.23 | 4.45 | 0.72 | 4.25 | 0.81 | 2.45 | 0.345 | 2.16 | 0.316 | 12.0 | 0.74 |
| D25 | 9.22 | 34.7 | 7.39 | 1.30 | 5.50 | 0.87 | 5.12 | 1.01 | 2.75 | 0.399 | 2.54 | 0.384 | 11.3 | 0.62 |
| Average | 8.19 | 30.3 | 5.84 | 1.21 | 4.71 | 0.73 | 4.34 | 0.83 | 2.39 | 0.347 | 2.26 | 0.340 | 11.6 | 0.71 |

[1] Values calculated from chondrite-normalized concentration. Chondrite values by Boynton (1984).

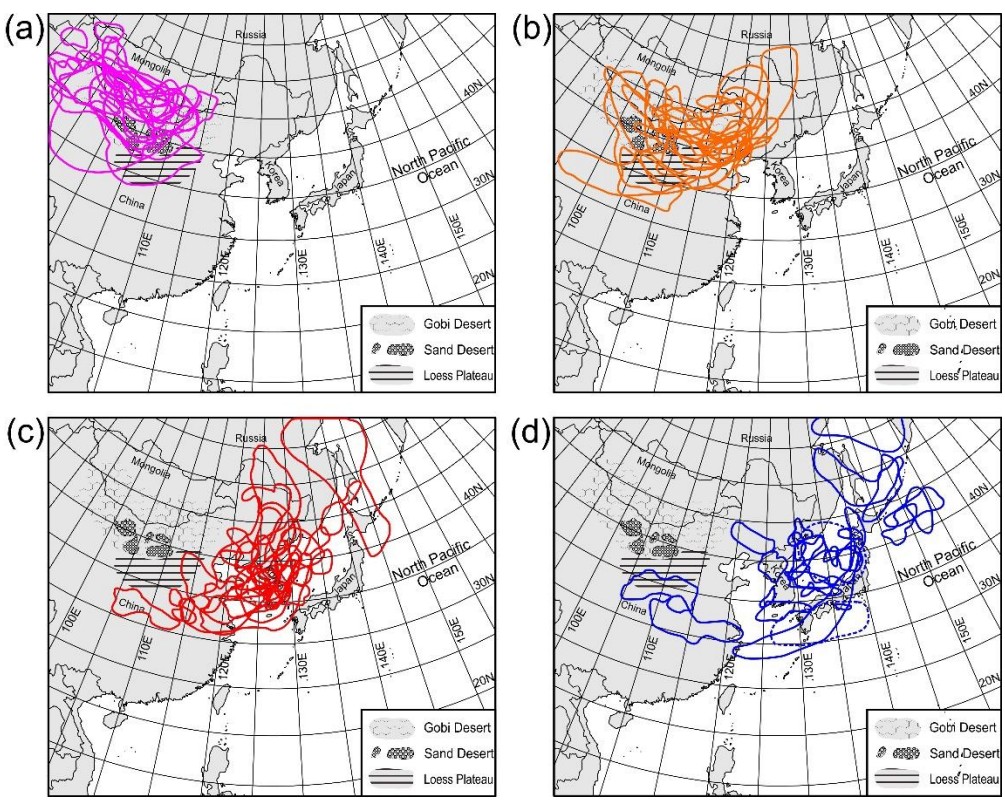

10   Figure 1. Dust storm outbreaks and migrations during 14 Asian dust events identified from the remote sensing
images obtained from the Communication, Ocean, and Meteorological Satellite (COMS) (2011–2018) and the
Multi-functional Transport Satellite-1R (MTSAT-1R) (2008–2010) satellites. The outbreak and migration path data
for individual dust events are provided in Supplementary Fig. 1. (a) The maximum extent of the dust event during
the storm outbreak. (b) Migration of dusty air toward the Korean Peninsula. (c) Dusty air crossing the Korean
15   Peninsula. (d) Migration of dusty air toward the North Pacific Ocean.

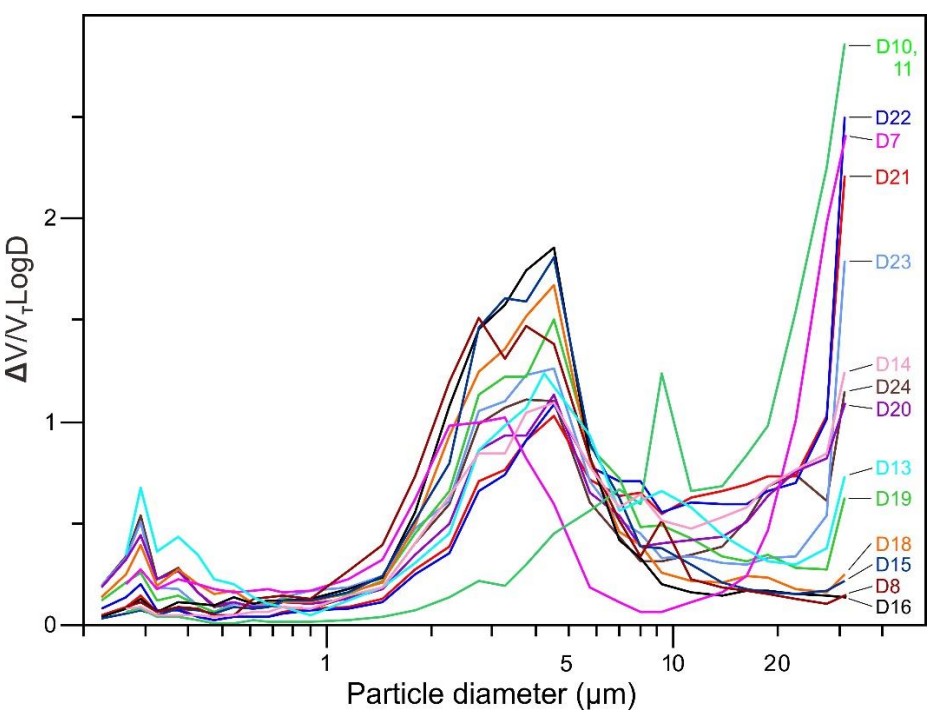

Figure 2. Volume-size distributions of aerosols during the Asian dust events, as measured by an optical particle counter located at the nearest Korea Meteorological Administration (KMA) monitoring stations. See Table 1 for the sampling site and time during the dust events.





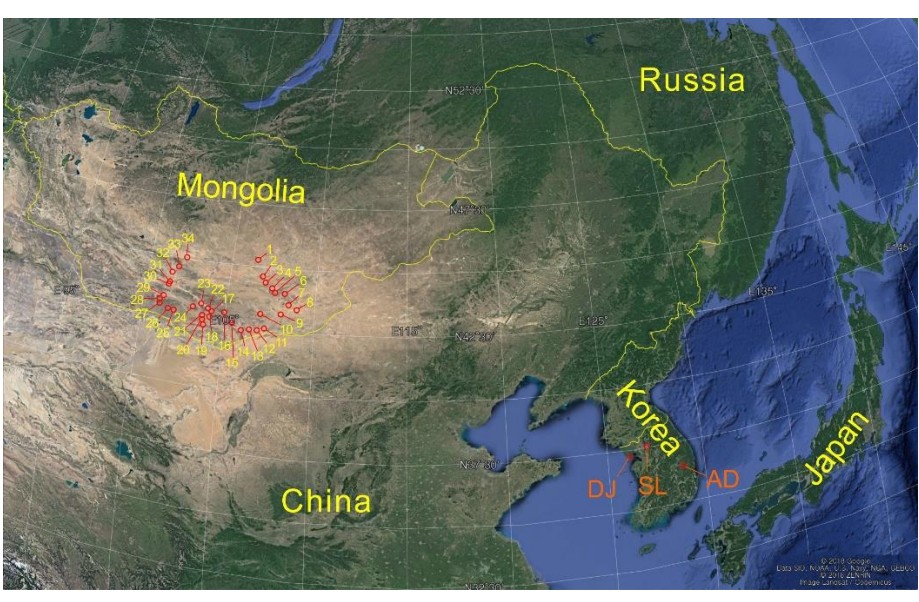

Figure 3. Three sites from which Asian dust samples were obtained in the Korean Peninsula, and the locations from which 34 soil samples were obtained in the Mongolian Gobi Desert. See Table 1 for the sampling site and time during the dust events. AD: Andong, DL: Deokjeok Island, SL: Seoul



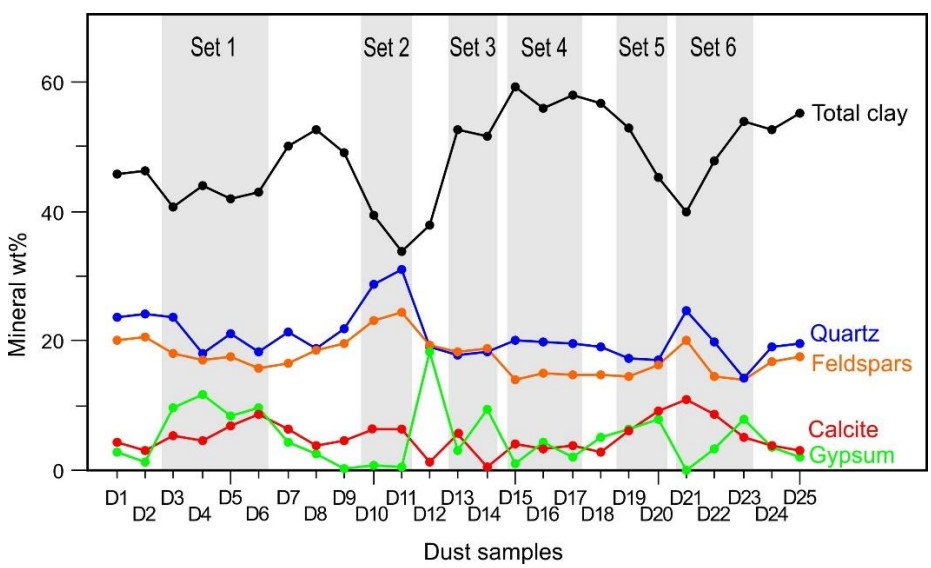

Figure 4. Time-series variation in the major-mineral contents of Asian dust, compared with the proxy (La/Yb)$_N$. Sample collection dates are provided in Table 1. Dust samples set 1, 5, and 6 comprise serial samples collected during individual dust events. Dust sample sets 2–4 comprise samples collected at different sites during individual dust events. A, amphibole; Ca, calcite; Ch, chlorite; G, gypsum; ISCMs, illite-smectite series clay minerals; Ka, kaolinite; Kf, K-feldspar; P, plagioclase; Q, quartz.

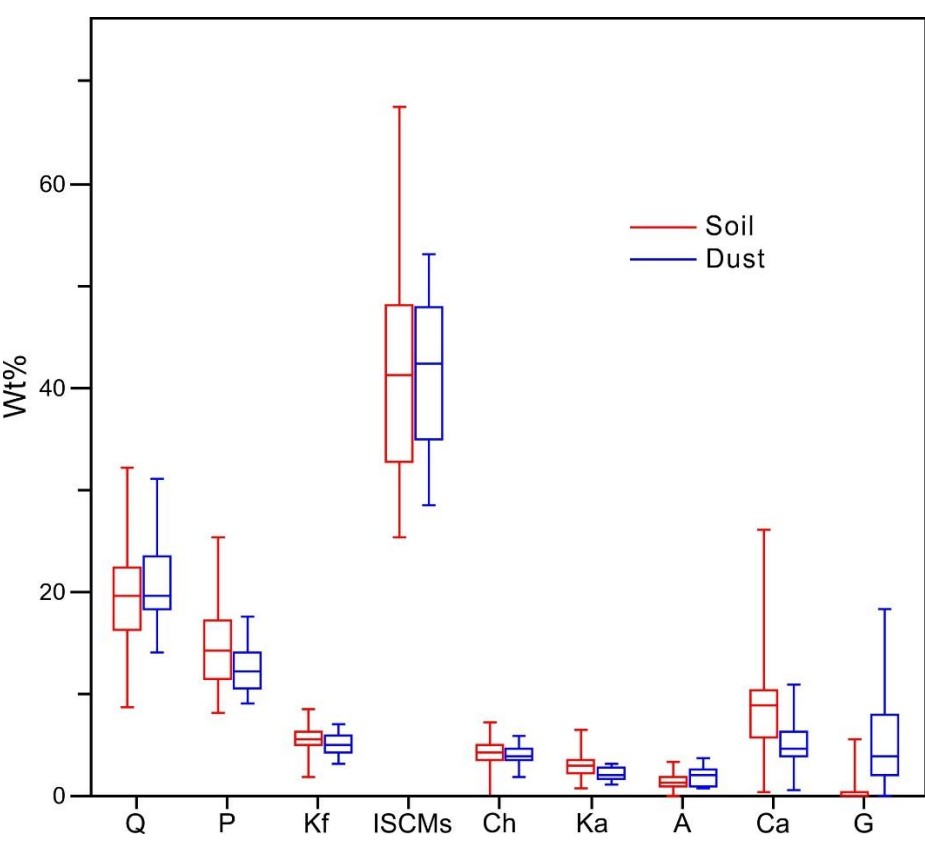

Figure 5. Box-whisker plot showing the mineral compositions of Asian dust and source soil samples. The top and bottom of the box define the third and first quartiles. The horizontal line in the center of the box is the second quartile, which is the median. The two ends of the vertical line crossing each box indicate minimum and maximum.



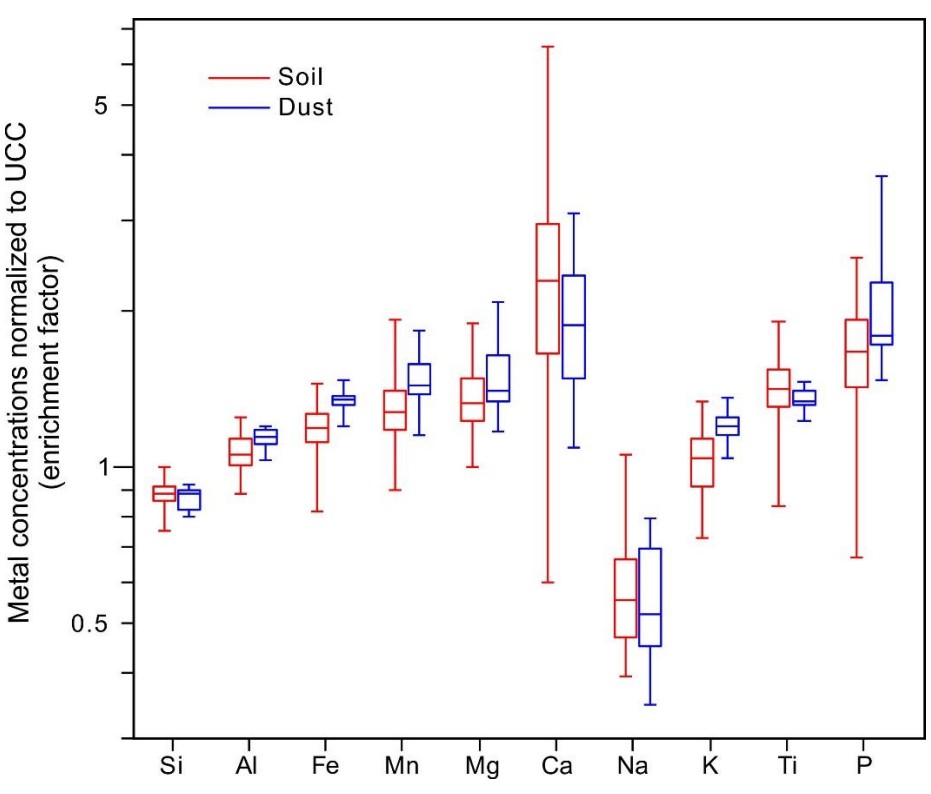

Figure 6. Box-whisker plot comparing the major element compositions between Asian dust and source soil samples, normalized to the average values of the upper continental crust (UCC) by Rudnick and Gao (2003).



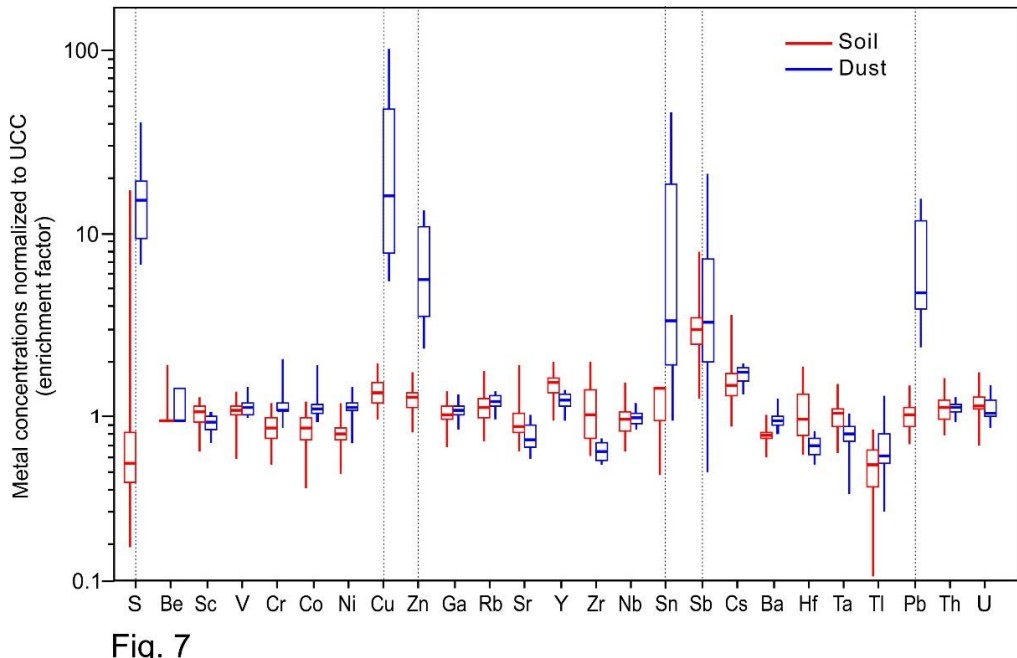

Fig. 7

Figure 7. Box-whisker plot comparing the trace element compositions between Asian dust and source soil samples, normalized to the average values of the UCC.

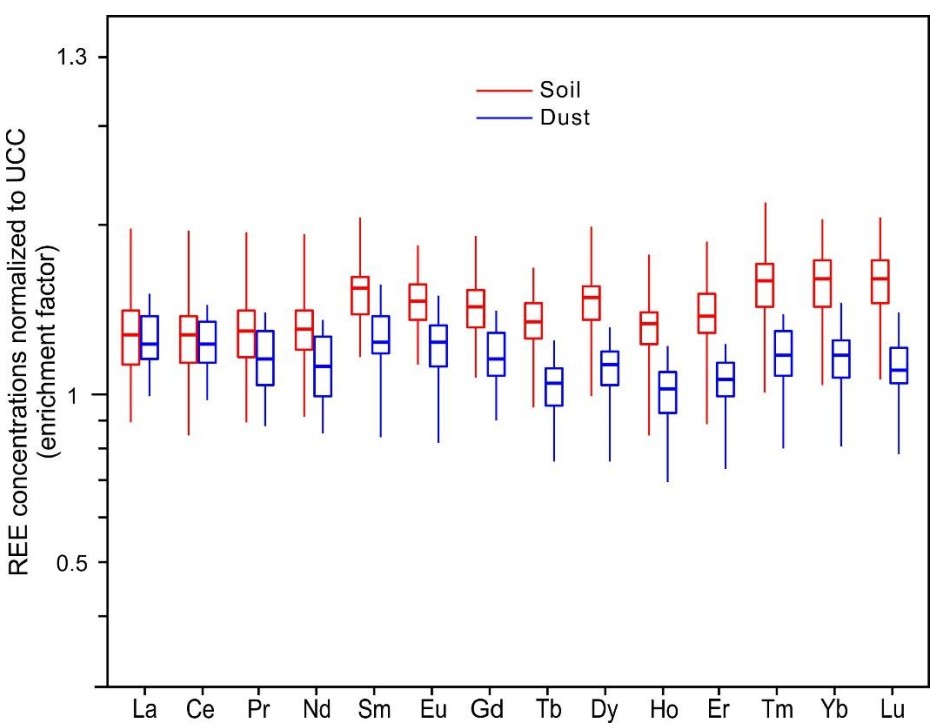

Figure 8. Box-whisker plot comparing the rare earth element compositions between Asian dust and source soil samples, normalized to the average values of the UCC.

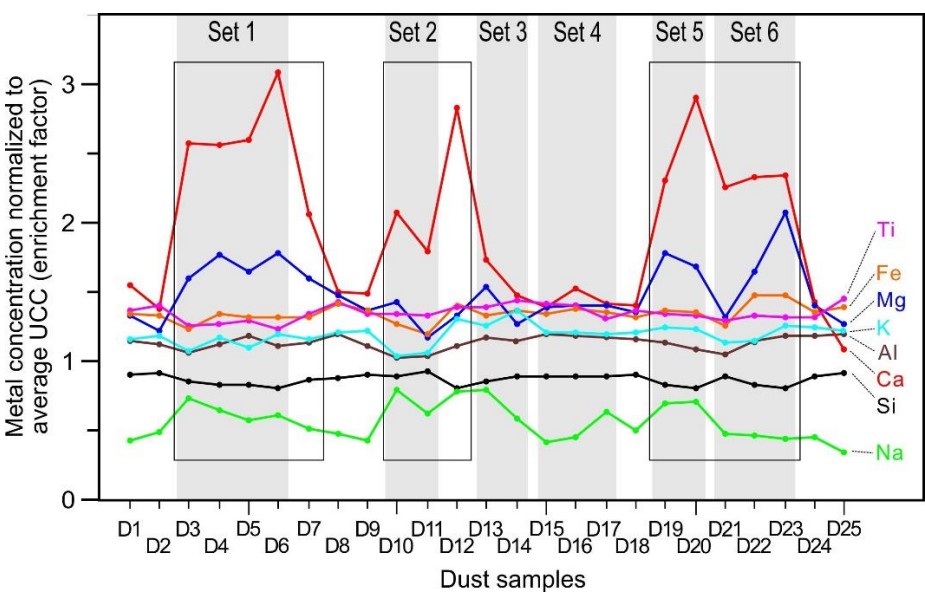

Figure 9. Time-series variation in major element compositions of Asian dust normalized to the average values of the UCC. Sample collection dates are provided in Table 1. Dust sample sets 1, 5, and 6 are serial samples collected during individual dust events. Dust sample sets 2–4 are samples collected at different sites during individual dust events. Dusts in square boxes were transported across the Chinese Loess Plateau and sandy deserts in northern China.





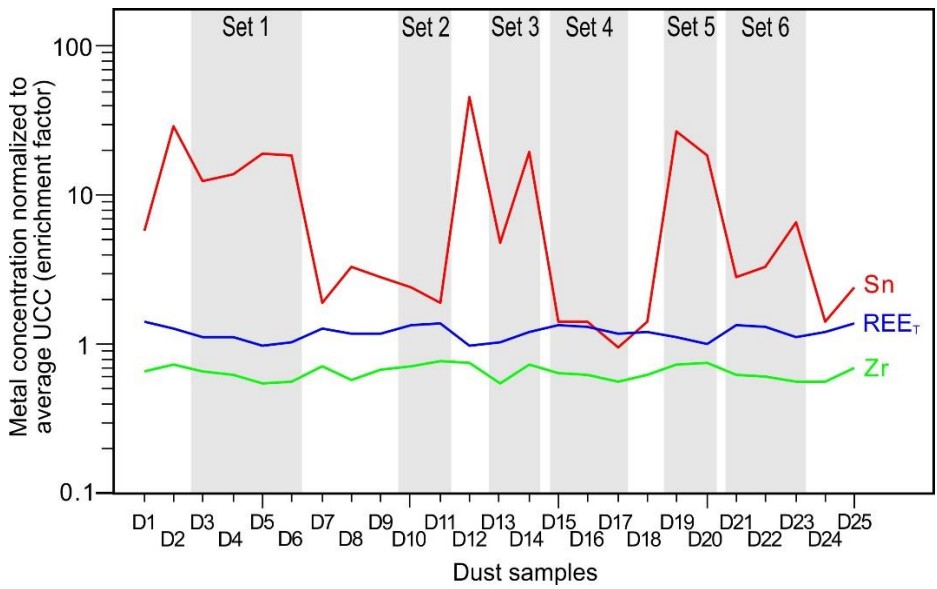

Figure 10. Time-series variation in trace element compositions of Asian dust normalized to the average values of the UCC.