# Peer review of "Mineralogy and geochemistry of Asian dust: Dependence on migration path, fractionation, and reactions with polluted air"

_Atmospheric Chemistry and Physics, 2019_

## Referee Comment (RC1) · Konrad Kandler (Referee) · 8 Feb 2020

Review of "Mineralogy and geochemistry of Asian dust: Dependence on migration path, fractionation, and reactions with polluted air" by Gi Young Jeong

The manuscript deals with the compositional properties of Asian dust outbreaks reaching the Korean peninsula. A time series reaching back more than a decade is analyzed in detail, proving a step towards a dust composition climatology, information highly useful for new generations of chemistry-aware aerosol and meteorology models. The author has done a careful and detailed analytical work of the samples, providing high-quality data set of dust geochemistry. Results are presented in a useful way, also for

further usage e.g. in modeling. The paper is well-written and concise. References are given where meaningful. I suggest the publication after some minor clarifications.

Remarks

General: When abbreviating mineral names, I suggest using common symbols (Kretz 1983).

Fig. 2 / size distributions: The Size distributions in particular towards the larger particles can be considerably biased by the type of inlet used for measurement. Please give some more details here. Also, Serno et al. refer to deposited aerosol from a sediment core. In principle, airborne size distributions should differ from deposited ones based on the same aerosol due to increasing deposition velocity for larger particles, so I'm not sure whether an agreement should be expected here.

Fig. 4: Where is the La/Yb proxy information in the plot, which is referred to in the caption?

Fig. 6: Is the Fe and K enrichment of dust versus soil significant?

Page 6 / line 7: What means 'side packing'?

Page 8 / line 16-17 and P9 / 22-23 and P13 / 18-21: The apparent anticorrelation between relative compositions might be misleading. If a major component – clay minerals here – variates, the other components must anticorrelate, a property of the normalized system. Feldspar and quartz show a similar temporal behaviour here, so a variation in clay minerals might drive the apparent anticorrelation; however, there is not proof just by composition data.

Page 14 / line 7-8: As dust emission might take its material mainly from the topmost crust (millimetres), couldn't the depletion of zircon have already happened before the emission, i.e. a zircon depletion from these top millimetres? From which depth were the soil samples collected?

Page 15 / line 24 onwards: What would happen actually to the dolomite in the acidic environment?

Page 16 / line 11-13: What could be the source of Sn?

Corrections

Page 2 / Line 5: Rephrase, maybe "Asia is one of the major mineral dust sources..."

P5 / 20: "shredded" -> "cut"

P11 / 21: "Taklamacan" -> "Taklamakan"

Is Section 5.4 a subsection of 5.3? Both are termed fractionation.

P15 / 5: Should that mean that the fractionation supposedly occurs on shorter distances? Please clarify the wording.

Kretz, R. (1983). "Symbols for rock-forming minerals." Am. Mineral. 68: 277-279.

---

## Author Comment (AC1) · 25 Feb 2020

Dear Dr. Kandler:

Thank you for your kind review of my manuscript.
I am pleased to reply to your constructive comments.

When abbreviating mineral names, I suggest using common symbols (Kretz 1983).
→ Re: Current abbreviations of mineral names will be replaced with symbols recommended by Kretz (1983) in revised manuscript

Fig. 2 / size distributions: The size distributions in particular towards the larger particles can be considerably biased by the type of inlet used for measurement. Please give some more details here. Also, Serno et al. refer to deposited aerosol from a sediment core. In principle, airborne size distributions should differ from deposited ones based on the same aerosol due to increasing deposition velocity for larger particles, so I'm not sure whether an agreement should be expected here.
→ Re: Particle-size data have been routinely measured using optical particle counters (GRIMM Aerosol Technik Model 180) at dust monitoring stations operated by the Korea Meteorological Administration. The instruments report particle numbers over 31 size bins from 0.25 to 32< μm. The specification sheet of the Model 180 states that "Sample air at a volume flow of 1.2 liter/minute is directly fed into the measuring cell by passing through a TSP (total suspended particles) head and the probe inlet", indicating that total suspended particles were directly fed into cell and measured. Details of the OPC will be added to the Section 2.1.2 in revised manuscript as follows: "OPC (GRIMM Aerosol Technik Model 180) reported particle numbers over 31 size bins from 0.25 to 32< μm. Sample air was directly fed into the measuring cell at a volume flow of 1.2 liter/minute by passing through a TSP head."
→ Re: The author agrees with reviewer. Size distribution of airborne dust may differ from deposited ones, not only by settling velocity dependent upon particle size but also by aggregation. It is required to compare the particle size of airborne dust particles with that of corresponding sediments at particular site remote from dust source. There are no systematic data yet. Nevertheless, the author experienced that the volume (or mass) size distribution is not so much different between airborne dust and corresponding sediments. For example, the median particle size of eolian sediments on Korean Peninsula deposited during the last glacial age (more intensive dust storm) was within the range of 5–6 μm (Jeong et al., 2013, Quaternary Science Reviews, 78, 283–300). The author likes to keep Serno et al. (2014) for comparison, but delete the over-interpreting sentence: "It is remarkable that particle size of dust is uniform from the western margin of the North Pacific Ocean to the subarctic mountains of the North America" from the section 2.1.2 in revised manuscript.

Fig. 4: Where is the La/Yb proxy information in the plot, which is referred to in the caption?
→ Re: It will be deleted in the revised manuscript.

Fig. 6: Is the Fe and K enrichment of dust versus soil significant?
→ Re: Data show that Fe and K are slightly enriched in dust relative to soil. It may be caused by the enrichment of fine mineral grains in dust such as K-bearing illitic clay minerals and iron oxides. However, it is more complicated by the presence of non-mineral K-bearing inorganic/organic aerosols and some Fe-rich pollutants emitted by fossil fuel combustion in East Asia. Thus, further experiments such as selective extraction are required in order to clarify the origin of Fe and K enrichment in dust samples on quantitative basis.

Page 6 / line 7: What means 'side packing'?
→ Re: Sample powders are normally loaded into the cavity of XRD holder and pressed with glass slide to obtain flat surface. This causes the preferred orientation of platy minerals hindering quantitative analyses. Careful packing of powders into the side of cavity covered with frost glass much reduces the preferred orientation. Moore and Reynolds (1997, X-ray Diffraction and the Identification and Analyses of Clay Minerals) will be added as a reference in revised manuscript.

Page 8 / line 16-17 and P9 / 22-23 and P13 / 18-21: The apparent anticorrelation between relative compositions might be misleading. If a major component – clay minerals here – variates, the other components must anticorrelate, a property of the normalized system. Feldspar and quartz show a similar temporal behaviour here, so a variation in clay minerals might drive the apparent anticorrelation; however, there is not proof just by composition data.
→ Re: The author agrees with reviewer. In revised manuscript, the author will delete "…, and showed a roughly inverse relationship with the total clay mineral content (R2=0.37)" in Page 8 / line 16-17, and "…, in the opposite direction to the clay mineral content" because of some statistical expression. However, the author like to keep P13 / 18-21.

Page 14 / line 7-8: As dust emission might take its material mainly from the topmost crust (millimetres), couldn't the depletion of zircon have already happened before the emission, i.e. a zircon depletion from these top millimetres? From which depth were the soil samples collected?
→ Re: Soil samples were collected from the exposed surface of Gobi Desert (several cm depth). The surface is generally covered with loose sediments. Sampling of several mm depth from the loose sediments is certainly challenging. Mineral fractionation is possible at the soil surface in addition to the possible fractionation during the long-range transport. In future field work, careful sampling could be carried out to confirm any mineralogical fractionation at the thin layer of surface.

Page 15 / line 24 onwards: What would happen actually to the dolomite in the acidic environment?
→ Re: This study could not confirm the fate of dolomite in the acidic atmospheric environment. Accuracy of quantitative XRD of clay-rich soils is generally much low in comparison to routine chemical analyses. It is difficult to state any trend from mineral compositions of around 1% quantity. Separate study focused on dolomite using electron microscopy may clarify the behavior of dust dolomite.

Page 16 / line 11-13: What could be the source of Sn?
→ Re: The source of Sn is fossil fuel combustion as written in Page 16 / line 8.

Page 2 / Line 5: Rephrase, maybe "Asia is one of the major mineral dust sources: : :"
→ Re: The phrase will be rearranged to "Asia is one of the major sources of mineral dust" in revised manuscript.

P5 / 20: "shredded" -> "cut"
→ Re: O.K.

P11 / 21: "Taklamacan" -> "Taklamakan"
→ Re: O.K.

Is Section 5.4 a subsection of 5.3? Both are termed fractionation.
→ Re: Section 5.3 deals with mineral fractionation, while Section 5.4 deals with chemical fractionation.

P15 / 5: Should that mean that the fractionation supposedly occurs on shorter distances? Please clarify the wording.
→ Re: 2000 km is generally long distance for mineral dust, while it is short distance compared with mineral dust crossed Pacific Ocean. Thus, the author did not use words such as 'short' or 'long'.

Sincerely

Gi Young Jeong
Department of Earth and Environmental Sciences
Andong National University

Andong 36729
Korea

---

## Referee Comment (RC2) · Anonymous Referee #2 · 8 Mar 2020

Jeong (2019) studied Asian dust transported to Korea for long term based on mineralogical and elemental compositions. Since there have been very many studies on the mineralogical and geochemical studies for Asian dust, I do not think that the novelty of this study is very high. However, there are several things as suggested below that can be improved based on the data given here, which can make the novelty higher than that in the present form. If it will be successful to increase the novelty, it might be possible to suggest that this manuscript is worth publishing in ACP.

(i) Source of the dust If you could clearly show that the Taklimakan Desert is not the source of the dust studied here based on the geochemical data, that would be impor-

tant. However, this study uses remote sensing data to distinguish the source, but the remote sensing data are not the main results of this study. There have been lots of REE data of sand in Taklimakan Desert (e.g., Honda et al., 2004), which may be useful to distinguish the source. If other data included in this study are also useful for this purpose, that would be also good.

(ii) Development of other tools Major element data can be more useful. For example, CIA (chemical index of alteration; Nesbitt and Young, 1982) may be good to show the clay mineral content based on the elemental data.

I have tried to plot Y/Ho vs. Zr/Hf (see the attached file), which shows systematic variation. These plots suggest the degree of influence of aqueous phase reactions. Generally speaking, phases formed in water (e.g., carbonate) is deviated from their chondritic values (Y/Ho = 28; Zr/Hf = 38; Bau, 1996). For example, these values decrease from non-chondritic value to more chondritic value in the order of D21 > D22 > D23 (see the data in the attached file). This may reflect decrease of carbonate content, which can be an important topic to discuss. Moreover, I think that this kind of information is related to the provenance of the dust, and the data of D21 to D23 may suggest the shift of the source during the event. Anyway, this kind of new data is strongly needed to make this work worth publishing.

Similarly, other trace element signatures should be tried in this study. I think that the plot of Cs/K vs. Rb/K may show the illite fraction among the whole minerals (Derkowski and McCarty, 2017), which is also effective to distinguish provenance of the materials.

(iii) Other minor comments (1) Sets 1, 5, and 6 and (ii) sets 2, 3, and 4 are different groups in terms of the grouping method. I think that it is not good idea to plot them into the same group. (2) Reaction of calcite with sulfuric acid to produce gypsum has been studied also by spectroscopic methods, which may be better to cite, since the method clearly reveal the process of the reaction in natural samples (Takahashi et al., 200). (3) L15 in P10: "ouliers" should be "outliers". (4) Table 1: Meaning of "Travel" is not clear.

(5) Table 4: Concentrations of Cu, Zn, and Pb are actually high in aerosols. Thus, low concentrations of these elements cannot be reason why the authors reported these elements only in a part of the samples.

References: Bau, M. Controls on the Fractionation of Isovalent Trace Elements in Magmatic and Aqueous Systems: Evidence from Y/Ho, Zr/Hf, and Lanthanide Tetrad Effect. Contrib. to Mineral. Petrol. 1996, 123, 323–333. Derkowski, A., & McCarty, D. K. Cesium, a water-incompatible, siloxane-complexed cation in Earth's upper crust. Geology 2017, 45, 899-902. Honda, M.; Yabuki, S.; Shimizu, H. Geochemical and Isotopic Studies of Aeolian Sediments in China. Sedimentology 2004, 51, 211-230. Nesbitt, H. W.; Young, G. M. Early Proterozoic Climates and Plate Motions Inferred from Major Elements Chemistry of Lutites. Nature 1982, 299, 715–717. Takahashi, Y.; Miyoshi, T.; Higashi, M.; Kamioka, H.; Kanai, Y. Neutralization of Calcite in Mineral Aerosols by Acidic Sulfur Species Collected in China and Japan Studied by Ca K-Edge X-Ray Absorption near-Edge Structure. Environ. Sci. Technol. 2009, 43, 6535-6540.

|     | Y    | Ho   | Y/Ho     | Zr  | Hf  | Zr/Hf    |
|-----|------|------|----------|-----|-----|----------|
| D1  | 27.6 | 0.93 | 29.67742 | 128 | 4   | 32       |
| D2  | 26.9 | 0.87 | 30.91954 | 139 | 3.7 | 37.56757 |
| D3  | 23.6 | 0.84 | 28.09524 | 128 | 3.9 | 32.82051 |
| D4  | 24.9 | 0.76 | 32.76316 | 119 | 3.4 | 35       |
| D5  | 21.3 | 0.72 | 29.58333 | 106 | 3.1 | 34.19355 |
| D6  | 23.2 | 0.76 | 30.52632 | 108 | 3   | 36       |
| D7  | 28.4 | 0.91 | 31.20879 | 137 | 3.6 | 38.05556 |
| D8  | 25.7 | 0.91 | 28.24176 | 111 | 3.8 | 29.21053 |
| D9  | 26.4 | 0.85 | 31.05882 | 129 | 4.2 | 30.71429 |
| D10 | 24.4 | 0.81 | 30.12346 | 136 | 3.9 | 34.87179 |
| D11 | 26.7 | 0.85 | 31.41176 | 148 | 4.4 | 33.63636 |
| D12 | 19.8 | 0.58 | 34.13793 | 144 | 4   | 36       |
| D13 | 22.4 | 0.71 | 31.5493  | 106 | 2.9 | 36.55172 |
| D14 | 26.5 | 0.87 | 30.45977 | 141 | 4.2 | 33.57143 |
| D15 | 29.7 | 0.97 | 30.61856 | 125 | 3.7 | 33.78378 |
| D16 | 28   | 0.92 | 30.43478 | 121 | 3.6 | 33.61111 |
| D17 | 26.6 | 0.88 | 30.22727 | 108 | 3.3 | 32.72727 |
| D18 | 27.7 | 0.93 | 29.78495 | 121 | 3.5 | 34.57143 |
| D19 | 24.9 | 0.77 | 32.33766 | 141 | 3.9 | 36.15385 |
| D20 | 21.8 | 0.71 | 30.70423 | 144 | 4.1 | 35.12195 |
| D21 | 25.2 | 0.77 | 32.72727 | 121 | 3.1 | 39.03226 |
| D22 | 26.6 | 0.88 | 30.22727 | 116 | 3.5 | 33.14286 |
| D23 | 23.8 | 0.83 | 28.6747  | 109 | 3.3 | 33.0303  |
| D24 | 25.7 | 0.81 | 31.7284  | 109 | 3.2 | 34.0625  |
| D25 | 28.8 | 1.01 | 28.51485 | 134 | 4   | 33.5     |

[Figure]

**Fig. 1.**

---

## Author Response (AR1)

MS No.: acp-2019-948
Title: Mineralogy and geochemistry of Asian dust: Dependence on migration path, fractionation, and reactions with polluted air

Authors: G. Y. Jeong,
MS Type: Research Article

I appreciate valuable comments by Konrad Kandler and anonymous referee during both the quick and ACPD reviews. I replied to all the comments and revised carefully the manuscript considering the comments by the referees.

**<Reply to the comments in Open Discussion>**

**Reply to the comments by Konrad Kandler**

**Comment:** When abbreviating mineral names, I suggest using common symbols (Kretz 1983).
**Reply**: Current abbreviations of mineral names will be replaced with symbols recommended by Kretz (1983) in revised manuscript

**Comment**: Fig. 2 / size distributions: The size distributions in particular towards the larger particles can be considerably biased by the type of inlet used for measurement. Please give some more details here. Also, Serno et al. refer to deposited aerosol from a sediment core. In principle, airborne size distributions should differ from deposited ones based on the same aerosol due to increasing deposition velocity for larger particles, so I'm not sure whether an agreement should be expected here.
**Reply(1)**: Particle-size data have been routinely measured using optical particle counters (GRIMM Aerosol Technik Model 180) at dust monitoring stations operated by the Korea Meteorological Administration. The instruments report particle numbers over 31 size bins from 0.25 to 32< μm. The specification sheet of the Model 180 states that "Sample air at a volume flow of 1.2 liter/minute is directly fed into the measuring cell by passing through a TSP (total suspended particles) head and the probe inlet", indicating that total suspended particles were directly fed into cell and measured. Details of the OPC will be added to the Section 2.1.2 in revised manuscript as follows: "OPC (GRIMM Aerosol Technik Model 180) reported particle numbers over 31 size bins from 0.25 to 32< μm. Sample air was directly fed into the measuring cell at a volume flow of 1.2 liter/minute by passing through a TSP head."
**Reply(2)**: The author agrees with reviewer. Size distribution of airborne dust may differ from deposited ones, not only by settling velocity dependent upon particle size but also by aggregation. It is required to compare the particle size of airborne dust particles with that of corresponding sediments at particular site remote from dust source. There are no systematic data yet. Nevertheless, the author experienced that the volume (or mass) size distribution is not so much different between airborne dust and corresponding sediments. For example, the median particle size of eolian sediments on Korean Peninsula deposited during the last glacial age (more intensive dust storm) was within the range of 5–6 μm (Jeong et al., 2013, Quaternary Science Reviews, 78, 283–300). The author likes to keep Serno et al. (2014) for comparison, but delete the over-interpreting sentence: "It is remarkable that particle size of dust is uniform from the western margin of the North Pacific Ocean to the subarctic mountains of the North America" from the section 2.1.2 in revised manuscript.

**Comment**: Fig. 4: Where is the La/Yb proxy information in the plot, which is referred to in the caption?
**Reply**: It will be deleted in the revised manuscript.

**Comment**: Fig. 6: Is the Fe and K enrichment of dust versus soil significant?

**Reply**: Data show that Fe and K are slightly enriched in dust relative to soil. It may be caused by the enrichment of fine mineral grains in dust such as K-bearing illitic clay minerals and iron oxides. However, it is more complicated by the presence of non-mineral K-bearing inorganic/organic aerosols and some Fe-rich pollutants emitted by fossil fuel combustion in East Asia. Thus, further experiments such as selective extraction are required in order to clarify the origin of Fe and K enrichment in dust samples on quantitative basis.

**Comment**: Page 6 / line 7: What means 'side packing'?
**Reply**: Sample powders are normally loaded into the cavity of XRD holder and pressed with glass slide to obtain flat surface. This causes the preferred orientation of platy minerals hindering quantitative analyses. Careful packing of powders into the side of cavity covered with frost glass much reduces the preferred orientation. Moore and Reynolds (1997, X-ray Diffraction and the Identification and Analyses of Clay Minerals) will be added as a reference in revised manuscript.

**Comment**: Page 8 / line 16-17 and P9 / 22-23 and P13 / 18-21: The apparent anticorrelation between relative compositions might be misleading. If a major component – clay minerals here – variates, the other components must anticorrelate, a property of the normalized system. Feldspar and quartz show a similar temporal behaviour here, so a variation in clay minerals might drive the apparent anticorrelation; however, there is not proof just by composition data.
**Reply**: The author agrees with reviewer. In revised manuscript, the author will delete "…, and showed a roughly inverse relationship with the total clay mineral content (R2=0.37)" in Page 8 / line 16-17, and "…, in the opposite direction to the clay mineral content" because of some statistical expression. However, the author like to keep P13 / 18-21.

**Comment**: Page 14 / line 7-8: As dust emission might take its material mainly from the topmost crust (millimetres), couldn't the depletion of zircon have already happened before the emission, i.e. a zircon depletion from these top millimetres? From which depth were the soil samples collected?
**Reply**: Soil samples were collected from the exposed surface of Gobi Desert (several cm depth). The surface is generally covered with loose sediments. Sampling of several mm depth from the loose sediments is certainly challenging. Mineral fractionation is possible at the soil surface in addition to the possible fractionation during the long-range transport. In future field work, careful sampling could be carried out to confirm any mineralogical fractionation at the thin layer of surface.

**Comment**: Page 15 / line 24 onwards: What would happen actually to the dolomite in the acidic environment?
**Reply**: This study could not confirm the fate of dolomite in the acidic atmospheric environment. Accuracy of quantitative XRD of clay-rich soils is generally much low in comparison to routine chemical analyses. It is difficult to state any trend from mineral compositions of around 1% quantity. Separate study focused on dolomite using electron microscopy may clarify the behavior of dust dolomite.

**Comment**: Page 16 / line 11-13: What could be the source of Sn?
**Reply**: The source of Sn is fossil fuel combustion as written in Page 16 / line 8.

**Comment**: Page 2 / Line 5: Rephrase, maybe "Asia is one of the major mineral dust sources: : :"
**Reply**: The phrase will be rearranged to "Asia is one of the major sources of mineral dust" in revised manuscript.

**Comment**: P5 / 20: "shredded" -> "cut"
**Reply**: O.K.

**Comment**: P11 / 21: "Taklamacan" -> "Taklamakan"
**Reply**: O.K.

**Comment**: Is Section 5.4 a subsection of 5.3? Both are termed fractionation.
**Reply**: Section 5.3 deals with mineral fractionation, while Section 5.4 deals with chemical fractionation.

**Comment**: P15 / 5: Should that mean that the fractionation supposedly occurs on shorter distances? Please clarify the wording.
**Reply**: 2000 km is generally long distance for mineral dust, while it is short distance compared with mineral dust crossed Pacific Ocean. Thus, the author did not use words such as 'short' or 'long'.

**<Reply to the comments by Anonymous referee>**

**Comment**: Jeong (2019) studied Asian dust transported to Korea for long term based on mineralogical and elemental compositions. Since there have been very many studies on the mineralogical and geochemical studies for Asian dust, I do not think that the novelty of this study is very high. However, there are several things as suggested below that can be improved based on the data given here, which can make the novelty higher than that in the present form. If it will be successful to increase the novelty, it might be possible to suggest that this manuscript is worth publishing in ACP.
**Reply**: One of the aims of this study was to present long-term data set of Asian dust and source soils obtained by consistent mineralogical and geochemical analyses. Although there have been lots of data on Asian dust in previous works, data are scattered and inconsistent each other due to a wide range of analytical methods, goal of study, sampling locations, and particle sizes. Long-term consistent data set would contribute to the multidisciplinary research of long-range transport dust. This study found the temporal/spatial variations of Asian dust properties which were interpreted with regard to transport path, reaction, and fractionation as well as the identification of dust source and a mineralogical and geochemical consistency between dust and source soils.

**Comment**: (i) Source of the dust if you could clearly show that the Taklimakan Desert is not the source of the dust studied here based on the geochemical data, that would be important. However, this study uses remote sensing data to distinguish the source, but the remote sensing data are not the main results of this study. There have been lots of REE data of sand in Taklimakan Desert (e.g., Honda et al., 2004), which may be useful to distinguish the source. If other data included in this study are also useful for this purpose, that would be also good.
(ii) Development of other tools Major element data can be more useful. For example, CIA (chemical index of alteration; Nesbitt and Young, 1982) may be good to show the clay mineral content based on the elemental data.
**Reply**: Geochemical comparison to source candidates is essential for tracking dust sources of ancient eolian deposits. In cases of present-day dust, satellite remote sensing provide clearly the outbreak and migration of Asian dust. However, geochemical data could confirm the dust source in combination with remote sensing data. I prepared three new diagrams: Th-Sc-La (Fig. 1), Th-Sc-Zr/10 (Fig. 2), and A-CN-K (Fig. 3) diagrams which will be added to revised manuscript. I also revised Fig. 6 in original manuscript (Fig. 4).

Geochemical data of fine fraction of Taklamakan Desert soils from previous works (Honda and Shimizu, 1998; Honda et al., 2004; Jiang and Yang, 2019) were plotted together with data of Mongolian Gobi Desert soils and Asian dust in this study (Figs. 1 and 2). Taklamakan soils (red circles) are distinguished from both the Asian dust (blue) and Mongolian Gobi soils (green). Asian dust matches with Gobi soils. Both the Mongolian Gobi soils and Asian dust were derived from the source rocks

formed in the tectonic setting of continental island arc, while Taklamakan soils were largely derived from source rocks formed in the passive margin (Figs 1 and 2).

The range of CIA values of Asian dust coincides with that of Gobi soils but it is distinguished from that of Taklamakan soils (Fig. 3). Fig. 3 indicate that the source rocks of Gobi soils and Asian dust are rather enriched with illitic clays in comparison to those of Taklamakan soils. My observation in the field and mineralogical analysis indicated that clay minerals in Gobi soils were not formed by the chemical weathering of rocks in the arid environment, but derived from Paleozoic to Cenozoic sedimentary rocks widespread in the Gobi Desert.

The major elements data of the Taklamakan Desert added on Fig. 6 of the original manuscript (Fig. 4). The Al, Fe, Ca, and K contents of the Taklamakan Desert soils greatly deviate from those of Asian dust and Mongolian Gobi soils, confirming that dust source is the Mongolian Gobi Desert (Fig. 4). The low Al, Fe, and K concentrations of the Taklamakan soils indicate low contents of clay minerals in comparison to those of Asian dust and Gobi soils, while high Ca concentration indicates the enrichment of calcite in the Taklamakan soils. Here is some limitation in using the geochemical data of the Taklamakan Desert by other works because of differences in size fractions of samples selected for analyses: Asian dust and Mongolian Gobi Desert soils in this study, < 20 μm; Taklamakan Desert soils in Jiang and Yang (2019), < 63 μm; Taklamakan Desert silts in Honda and Shimizu (1998), < 45 μm. Nevertheless, my geochemical data of < 63 μm fraction in Mongolian Gobi soils showed a consistency with the data of < 20 μm fractions (Fig. 5).

[Figure]

Fig. 1. Th–Sc–La plot

[Figure]

Fig. 2. Th–Sc–Zr/10 plot

[Figure]

Fig. 3. A–CN–K plot

[Figure]

Fig. 4. Major elements concentrations.

[Figure]

Fig. 5. Th–Sc–La plot. Comparison of 20 and 60 μm fractions

**Comment**: I have tried to plot Y/Ho vs. Zr/Hf (see the attached .le), which shows systematic variation. These plots suggest the degree of influence of aqueous phase reactions. Generally speaking, phases formed in water (e.g., carbonate) is deviated from their chondritic values (Y/Ho = 28; Zr/Hf = 38; Bau, 1996). For example, these values decrease from non-chondritic value to more chondritic value in the order of D21 > D22 > D23 (see the data in the attached file). This may reflect decrease of carbonate content, which can be an important topic to discuss. Moreover, I think that this kind of information is related to the provenance of the dust, and the data of D21 to D23 may suggest the shift of the source during the event. Anyway, this kind of new data is strongly needed to make this work worth publishing.

**Reply(1)**: Data are plotted on the Y/Ho-Zr/Hf diagram (Bau, 1996) (Figs. 6 and 7). Bau (1996) defined CHARAC (CHArge-and-RAdius-Controlled) field of rocks preserving the Y/Ho and Zr/Hf ratios of chondrite. Fig. 6 shows that all the data of Asian dust, Gobi and Taklamakan soils fall into CHARAC field, indicating little fractionation. Fig. 7 shows that the Y/Ho and Zr/Hf domain of water and water-related minerals (Bau, 1996) are located far from CHARAC fields. Thus, it is difficult to discuss the variation of Y/Ho and Zr/Hf ratios of Asian dust in relation to the fractionation caused by aqueous reaction in the dust source.

[Figure]

Fig. 6.

[Figure]

Fig. 7

**Reply(2)**: Although calcite content decreased from D21 to D23 with time in single dust event, it is certainly due to calcite-gypsum conversion rather than changing calcite content in pristine dust or changing provenance. Ca contents of samples D21, D22, and D23 were not changed as shown in Table below (also Table 3 in manuscript).

| wt% | D21 | D22 | D23 |
|---|---|---|---|
| Ca | 5.8 | 5.98 | 6.02 |
| Calcite | 11.0 | 8.5 | 5.2 |
| Gypsum | 0.0 | 3.3 | 7.9 |

The decrease of Y/Ho from 32.7 (D21) to 28.7 (D23) during the same dust event may be related to the change of mineral composition by particle-size change. Quartz and feldspar contents decreased with increasing clay content from the first sample (D21) to the last sample (D23). The particle size of sample D23 suspended in air for longer time decreased within (shown in Fig. 2 in manuscript), probably resulting in the selective removal of some minor heavy minerals. However, it is difficult to assign Y and Ho to specific minerals.

**Comment**: Similarly, other trace element signatures should be tried in this study. I think that the plot of Cs/K vs. Rb/K may show the illite fraction among the whole minerals (Derkowski and McCarty, 2017), which is also effective to distinguish provenance of the materials.

**Reply**: Cs/K and Cs/Rb were normalized using compositions of upper continental crust (UCC) (Fig. 8). Fine fractions of Taklamakan soils are clearly discriminated from both the Mongolian Gobi Desert and Asian dust, supporting that Asian dust was originated in Mongolian Gobi Desert. Derkowski and McCarty (2017) showed that Cs is almost partitioned to illite-smectite minerals. Fig. 8 suggest that fine fractions of Mongolian Gobi soils are enriched in illitic clay minerals relative to those of Taklamakan Desert soils.

[Figure]

Fig. 8

**Referee**: (iii) Other minor comments

(1) Sets 1, 5, and 6 and (ii) sets 2, 3, and 4 are different groups in terms of the grouping method. I think that it is not good idea to plot them into the same group.

**Reply**: I agree. It is not good to put both groups into one diagram. I have tried several alternatives using averages of sets 2, 3, and 4, or vertical alignment, but been more complicated. I hope to keep current setting by changing the shade of sets 2, 3, and 4 in Figs. 4, 9, and 10.

(2) Reaction of calcite with sulfuric acid to produce gypsum has been studied also by spectroscopic methods, which may be better to cite, since the method clearly reveal the process of the reaction in natural samples (Takahashi et al., 200).
**Reply**: Takahashi et al. (2009, 2014) will be cited in the revised version.

(3) L15 in P10: "ouliers" should be "outliers".
**Reply**: O.K.

(4) Table 1: Meaning of "Travel" is not clear.
**Reply**: "Travel" was replaced by "Transport"

(5) Table 4: Concentrations of Cu, Zn, and Pb are actually high in aerosols. Thus, low concentrations of these elements cannot be reason why the authors reported these elements only in a part of the samples.
**Reply**: Mass of some dust samples were very small and not enough to analyze full range of elements. Thus, analyses of Cu, Pb, and Zn were not done for several samples.

<Preparation of revised manuscript>

→ One new Figure 11 (including Fig. 1, 2, 3, and 8 discussed above) will be added on manuscript.
→ Figs. 4, 6, 9, and 10 in manuscript will be revised.
→ Fig. 5 above will be added on supplementary file as Fig. S3.
→ Tables 1, 3, S2 will be revised
→ Text will be revised also.

**<Corrected and added parts of the manuscript are shown in next pages>**

[revised manuscript text omitted]

---

## Author Response (AR2)

Dear Professor Ryan Sullivan:

Thank you for your kind review of my manuscript.
I adjusted the sizes of boxes in Fig. 6 to the same size.

Sincerely

Gi Young Jeong